# Differential neuronal survival defines a novel axis of sexual dimorphism in the *Drosophila* brain

## Graphical abstract

## Highlights

- Systematic analysis of sex-specific transcriptomic diversity in the adult brain

- *doublesex* and *fruitless* explain the majority of sex differences in the adult brain

- Links *doublesex* and *fruitless* transcriptomic diversity to neuron anatomy

- Birth order represents a novel axis of sexual differentiation in the central brain

## Authors

Aaron M. Allen, Megan C. Neville, Tetsuya Nojima, Faredin Alejevski, Stephen F. Goodwin

## Correspondence

aaron.allen@cncb.ox.ac.uk (A.M.A.), stephen.goodwin@cncb.ox.ac.uk (S.F.G.)

## In brief

Allen et al. leverage a high-resolution single-cell transcriptomic atlas of the adult *Drosophila* central brain to map sexual dimorphism. They show that dimorphic neuronal types correlate with the expression of the sex-determination transcription factors Doublesex and Fruitless. The authors uncover extensive transcriptional diversity within these dimorphic cell types and link their molecular profiles to anatomical identities. They further identify birth order as a previously unrecognized axis of dimorphism, driven by sex-biased neuronal survival.

 Allen et al., 2026, Cell Genomics 6, 101125
March 11, 2026 © 2025 The Authors. Published by Elsevier Inc.

CellPress

**Article**

# Differential neuronal survival defines a novel axis of sexual dimorphism in the *Drosophila* brain

Aaron M. Allen,[1,2,*] Megan C. Neville,[1,2] Tetsuya Nojima,[1] Faredin Alejevski,[1] and Stephen F. Goodwin[1,3,*]

[1]Centre for Neural Circuits and Behaviour, University of Oxford, Oxford OX1 3TA, UK
[2]These authors contributed equally
[3]Lead contact
*Correspondence: aaron.allen@cncb.ox.ac.uk (A.M.A.), stephen.goodwin@cncb.ox.ac.uk (S.F.G.)

## SUMMARY

Sex differences in behaviors arise from variations in female and male nervous systems, yet the cellular and molecular bases of these differences remain poorly defined. Here, we employ an unbiased, single-cell transcriptomic approach to investigate how sex influences the adult *Drosophila melanogaster* brain. We demonstrate that sex differences do not result from large-scale transcriptional reprogramming, but rather from selective modifications within shared developmental lineages mediated by the sex-differentiating transcription factors Doublesex and Fruitless. We reveal, with unprecedented resolution, the extraordinary genetic diversity within these sexually dimorphic cell types and find that birth order represents a novel axis of sexual differentiation. Neuronal identity in the adult reflects spatiotemporal patterning and sex-specific survival, with female-biased neurons emerging early and male-biased neurons arising later. This pattern reframes dimorphic neurons as "paralogous" rather than "orthologous," suggesting sex leverages distinct developmental windows to build behavioral circuits, and highlights a role for exaptation in diversifying the brain.

## INTRODUCTION

The nervous systems of sexually reproducing species diverge between the sexes to produce distinct behavioral repertoires.[1] As neuronal diversification in animals relies on the intrinsic mechanisms of spatial and temporal patterning, each sex must act differentially upon these mechanisms to establish neural circuits that meet its own behavioral needs.[2] Understanding how chromosomal sex informs the relationships between transcriptomic signatures and specific anatomical, physiological, and functional properties of cells in the nervous system is critical to understanding sexual behavior.

The vinegar fly, *Drosophila melanogaster*, provides an ideal model for exploring these questions due to its compact yet highly specialized central nervous system, well-characterized sex-determination pathway, and rich repertoire of sexually dimorphic behaviors. In *Drosophila*, sex is determined early in embryogenesis via an X-chromosome-counting mechanism, which activates the master regulator *Sex-lethal* (*Sxl*) in females.[3–5] *Sxl* triggers a hierarchical alternative splicing cascade via *transformer* (*tra*) that culminates in the expression of two key transcription factors, Doublesex (Dsx) and Fruitless (Fru), which initiate many irreversible cell-autonomous sexual differentiation events.[6,7]

The expression of Dsx and Fru in the nervous system begins post-embryogenesis, sculpting sex-specific neural circuits required for adult sexual behavior.[8–11] Neurons in *Drosophila* originate from neural stem cells, known as neuroblasts, which divide to generate paired hemilineages. These hemilineages represent the fundamental developmental and anatomical units of the central nervous system.[12–17] The selective expression of Dsx and Fru within specific hemilineages governs the sex-specific survival and connectivity of dimorphic neurons, laying the groundwork for innate sexual behaviors.[6] However, the precise transcriptional landscape underlying the developmental specification of individual dimorphic neurons remains unclear.

Existing single-cell RNA sequencing (scRNA-seq) atlases have provided valuable overviews of cellular diversity across *Drosophila* tissues.[18] The central brain is an elaborately complex tissue that is critical for integrating sensory information and internal state to drive behavioral outputs. To date, no single study has achieved sufficient cellular depth to detect the breadth of dimorphisms between the sexes across the brain. To overcome this limitation, we generated a high-depth, sexed single-cell transcriptomic atlas of the adult *Drosophila* central brain. By integrating our newly generated sex-specific data with multiple publicly available scRNA-seq datasets, we constructed a comprehensive meta-central brain neuron atlas.[19] In our companion article, we show that neuronal identity is primarily defined by hemilineage and secondarily by birth order within a hemilineage.[19] Here, we build on this framework to investigate how sex shapes neuronal identity across the central brain, revealing its influence at unprecedented resolution.

Our analysis reveals that sexually dimorphic neuronal populations overwhelmingly correlate with the expression of *dsx* and

*fru*, reinforcing their roles as key determinants of neuronal sexual identity. We generated *dsx*[+] and *fru*[+] neuron subatlases, enabling us to resolve distinct transcriptomic cell types, which we linked to anatomically defined cell types using genetic intersectional tools, allowing an understanding of transcriptomic diversity in a spatial and functional context. Our data uncover a novel relationship between sex and developmental birth order within hemilineage: female-biased neurons are typically born earlier, while male-biased neurons are typically born later. This finding suggests that sexually dimorphic neurons within a hemilineage often arise through developmental shifts in the survival of cells distinguished by their transcriptional identities and birth dates rather than through a one-to-one correspondence of type between female and male counterparts, with females simply having fewer cells of that type. This discovery reconceptualizes traditional models of sexual differentiation and provides a new framework for understanding the evolution of sexually dimorphic neural circuits.

## RESULTS

### Sex-specific transcriptomic diversity in the adult *Drosophila* central brain

To investigate how chromosomal sex shapes transcriptional diversity in the nervous system, we generated a sexed single-cell transcriptomic atlas of the *D. melanogaster* adult central brain.[19] We integrated our data with multiple independent central-brain-containing single-cell datasets, resulting in a comprehensive meta-central-brain neuronal atlas comprising 329,466 neurons, achieving a cellular depth of coverage of 9.8× (Figure 1A; see STAR Methods and Allen et al.[19]). Using established marker genes, we assigned broad cellular identities to 246 transcriptionally distinct neuronal cell types, many of which encompass multiple transcriptionally and functionally distinct subtypes. In our companion study, we demonstrated that adult neuronal transcriptional identity overwhelmingly reflects developmental origin, based on shared hemilineage identity (neurons produced from one branch of a neuroblast lineage) and temporal windows of neurogenesis.[19] Our adult brain atlas provides a framework to explore how sex acts upon a shared developmental scaffold to generate dimorphism.

To explore sex differences in transcriptionally distinct cell types within central brain neurons, we first examined the expression of genes involved in the sex-determination hierarchy (SDH) and dosage compensation (Figure 1B). The SDH governs somatic sexual differentiation, and most of its members were ubiquitously expressed across all neuronal cell types in both sexes (Figure 1C), including the master regulator *Sxl*, its translational repressor *sister-of-Sex-lethal*[20–22] (*ssx*), and the splicing factors *tra* and *transformer-2* (*tra2*), whose widespread expression has recently been confirmed throughout the body.[23] In contrast, the transcription factors *dsx* and *fru* exhibited cell-type-restricted expression, marking the first level of spatially specific transcriptional regulation underlying sexual differentiation in the central brain (Figure 1C). Analysis of dosage-compensation genes (Figure 1C) revealed that male-specific *lncRNA:roX1* and *lncRNA:roX2* were ubiquitously expressed in males, while other dosage-compensation genes were ubiquitously detected in both

sexes (Figures S1A and S1B). Several SDH and dosage-compensation genes exhibited sex-biased expression across all neuronal cell types: *Sxl* and *tra* displayed significant female-biased expression, whereas *ssx* and *msl-2* showed significant male-biased expression (Figure 1D). Notably, three genes—*Sxl*, *tra*, and *msl-2*—were more highly expressed in the sex that produces the corresponding functional protein product (Figures 1B and 1D).

### Differential gene expression between the sexes in central brain neurons

When examining sex-biased gene expression across the central brain in each neuronal cell type (Figure 1E), we identified a total of 114 differentially expressed genes (DEGs); 74 were female biased and 47 were male biased, but 7 of these 121 genes were female biased in one cell type and male biased in another (Figures 1E, 1F, S1A–S1I, and S2; see STAR Methods). We validated the robustness of this bias across all datasets within our meta-atlas, excluding genes whose sex bias was confined to individual datasets (Figures S2 and S3D). Most genes (87%) were differentially expressed in only a single cell type, although the specific cell type varied for each gene. The low number of sex-biased genes identified in the central brain (~1% of expressed genes), along with the modest magnitude of their difference, is striking, especially when compared to other tissues.[18] Gene Ontology enrichment analysis revealed significant overrepresentation of terms related to transmembrane signaling, cyclic-nucleotide-mediated pathways, and behaviors associated with mating and courtship (Figure S3A), suggesting sex differences may influence circuit-level properties such as neuronal excitability, synaptic plasticity, and hormonal responsiveness. Our data support the prevailing model that sex influences the transcriptome of mature neurons through modest, cell-type-specific tuning within a largely shared framework rather than through broad transcriptional reprogramming across the brain.[10,18,24–35] This, however, does not rule out the possibility that more extensive sex-specific transcriptional differences emerge earlier during development.

Analysis of the genomic distribution of DEGs revealed a marked enrichment of sex-biased transcripts on the X chromosome relative to individual autosomal arms, with this bias being especially pronounced among female-biased genes (Figure 1G). This pattern, consistent with prior observations,[18,36–38] supports the notion that the X chromosome serves as a favorable genomic context for the emergence and regulation of sex-biased gene expression.[36–38] In male *Drosophila*, global dosage-compensation mechanisms elevate X-linked gene expression to approximate that of the two X chromosomes in females. Interestingly, we identified *Myc*, a broadly female-biased transcription factor residing on the X chromosome, which appears to escape dosage compensation in males and has been implicated in the activation of *Sxl* in females (Figures S3B–S3D).[5,39,40]

Differential expression does not need to be large to exert significant biological effects, and small but widespread sex-biased differences may accumulate to generate significant dimorphic traits.[41] When we lowered our thresholds to capture all DEGs with any fold change, we observed numerous ubiquitously expressed genes showing subtle yet consistent sex-biased expression across multiple neuronal populations (Figures S3B

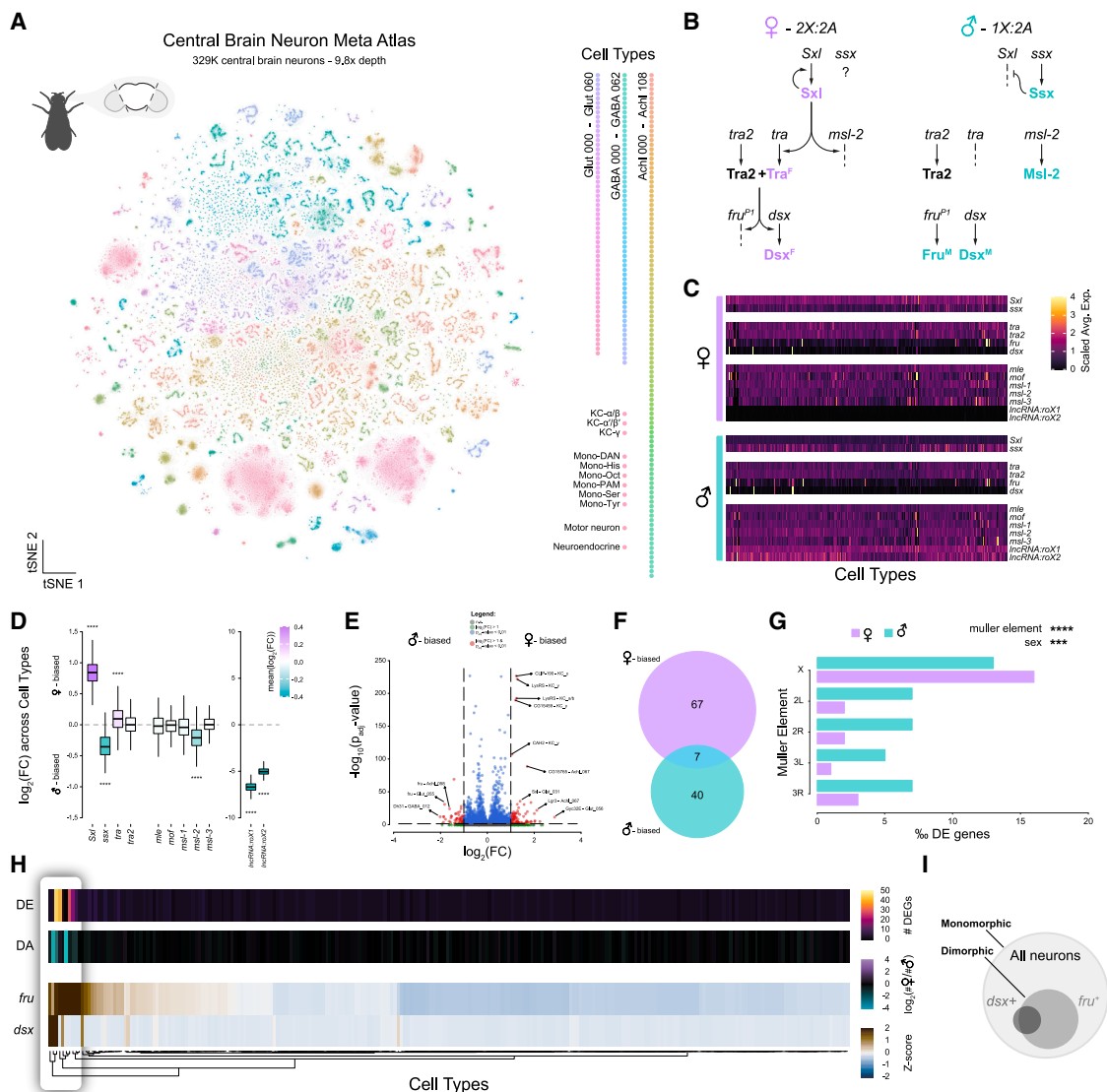

**Figure 1. Sexual dimorphism in *Drosophila* central brain at single-cell resolution**

(A) tSNE of 329,466 *Drosophila* central brain neurons, representing 9.8-times depth coverage. Cell types are defined by unique colors. Data are from our companion paper.[19]

(B) Schematic of the sex-determination dosage compensation hierarchy in *Drosophila melanogaster*.

(C) Heatmap of scaled average expression of sex-determination and dosage-compensation genes across cell types of the central brain in females (top) and males (bottom).

(D) Boxplots showing log₂ fold change (FC) of sex-biased sex-determination (left) and dosage-compensation (right) gene expression across all cell types in the central brain. Bonferroni-adjusted chi-squared ****$p < 0.0001$.

(E) Volcano plot of differentially expressed (DE) genes exhibiting male-biased (left) and female-biased (right) expression identified in all neuronal cell types across the central brain. Dashed lines indicate statistical significance thresholds.

(F) Euler diagram representing the number of DE genes among cell types unique to females or males or found in both sexes (albeit in different cell types). $p_{adj} < 0.01$, $\log_2(FC) > 1$.

(G) The distribution of DE genes across Muller elements between females and males. Bonferroni-adjusted chi-squared ***$p < 0.001$ and ****$p < 0.0001$.

(H) Heatmaps of the number of DE genes (top), differential abundance of cell numbers between the sexes (DA, middle), and Z-scored expression of *fru* and *dsx* (bottom) across all cells.

(I) Central brain neurons expressing *dsx* or *fru* exhibit transcriptional dimorphism; all other neurons are classified as monomorphic.

and S3C). The broad expression of many SDH genes throughout the central brain (Figure 1C) indicates that genes such as *Sxl* and *tra* may influence widespread, low-level patterns of sex-biased expression independently of *dsx* and *fru*, as has been reported in other tissues.[23,42,43] Intriguingly, among the ubiquitously sex-biased genes, a pronounced pattern emerged, with the

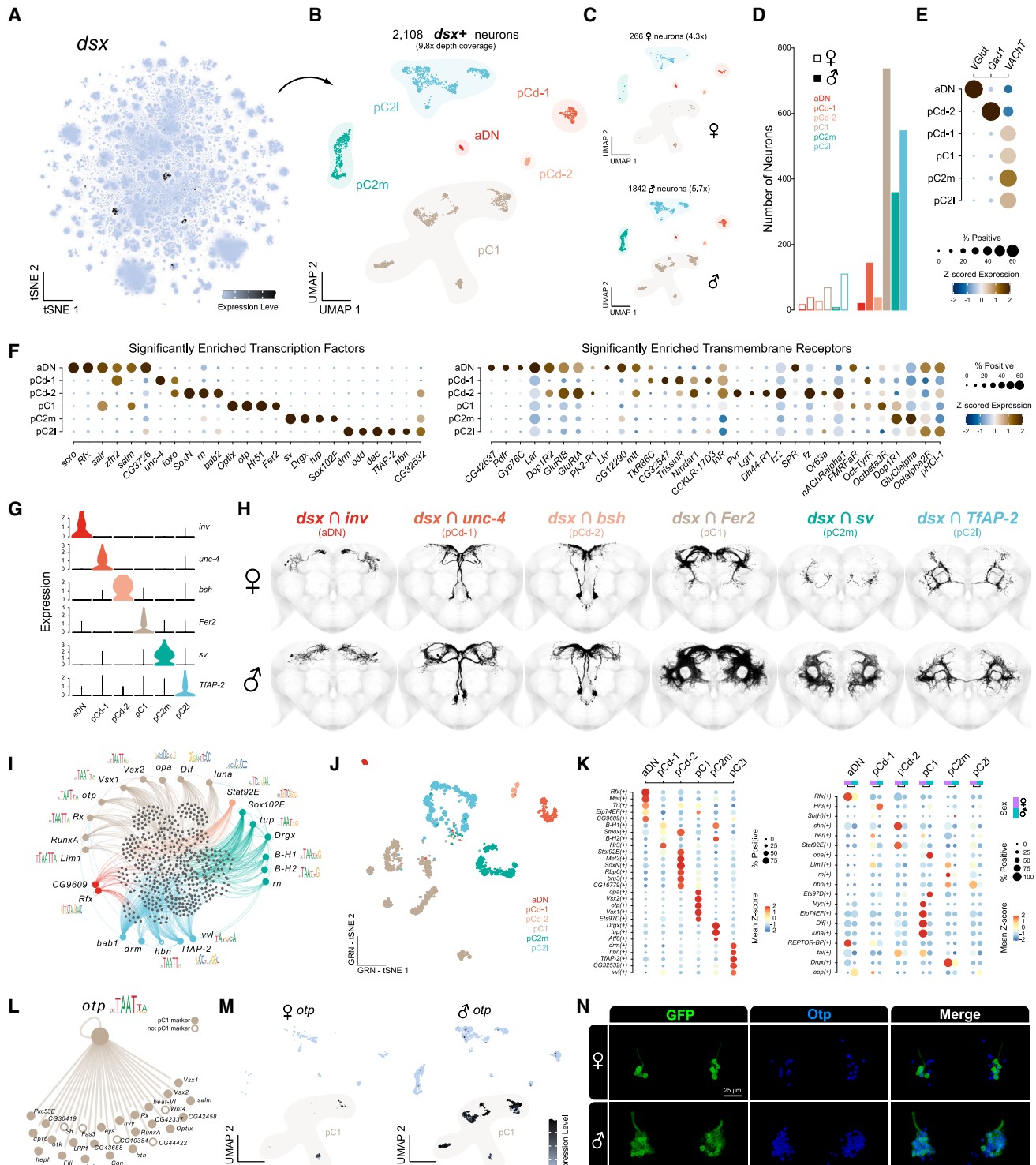

**Figure 2. Transcriptional diversity of *doublesex* neurons in the central brain**

(A) tSNE of *dsx* expression across central brain neurons.

(B) Uniform manifold approximation and projection (UMAP) of subclustered *dsx*⁺ neurons, with transcriptionally defined cell types annotated.

(C) UMAPs of *dsx*⁺ neurons in females (top) and males (bottom).

(D) Bar chart of the number of *dsx*⁺ neurons across transcriptionally defined central brain subtypes between females and males.

(E) Dot plot showing expression of key biomarkers for acetylcholine (*VAChT*), glutamate (*VGlut*), and GABA (*Gad1*) across *dsx*⁺ neuronal cell types.

*(legend continued on next page)*

majority of ribosomal (129/164) and mitochondrial (81/89) genes trending male biased, suggesting potential sex-specific metabolic demands in neurons (Figure S3E). However, these trends approach the detection limits of our approach and may be influenced by technical factors.

Finally, we investigated the distribution of sex differences across neuronal populations of the central brain. We considered two measures: the number of DEGs per cell type (using stringent filtering) and the differential abundance (the number of female cells relative to male cells) for each cell type (Figures 1H and S1J). Most cell types showed little to no difference in these metrics. A small number of cell types, however, exhibited dramatic differences in the number of genes differentially expressed and the differential abundance between the sexes. When exploring what distinguishes these cell types, we found they showed strong enrichment for *dsx* or *fru*. These findings suggest that the strongest sex differences in the brain are highly regional, with *dsx* and *fru* serving as robust markers of neuronal populations where sex differences are most pronounced, consistent with their well-established roles in sexual differentiation.[6] Accordingly, we refer to *dsx*+ and *fru*+ neurons as the transcriptionally sexually dimorphic contingent of the central brain and to neurons lacking expression of both genes as transcriptionally sexually monomorphic (Figure 1I), while acknowledging that low-level and widespread differences are present throughout the brain.

### Distinct transcriptional signatures define *dsx*+ neuronal cell types

Next, we explored the transcriptional diversity of *dsx*-expressing neurons. We extracted and subclustered all *dsx*+ cells from our meta-central brain neuron atlas (Figure 2A). The resulting *dsx*+ atlas, representing 9.8-fold depth of coverage, revealed six broad transcriptionally distinct neuronal subpopulations in both sexes (Figures 2B and 2C). To correlate these subpopulations with anatomically defined, lineage-specific *dsx*+ cell types in the central brain, we leveraged established cell number representation and neurotransmitter usage to predict their identities (Figures 2D and 2E). Further molecular characterization revealed cell-type-specific expression of transcription factors and transmembrane receptors, indicating that distinct regulatory and physiological profiles underlie sex-specific circuit functions (Figure 2F).

To validate predicted identities, we employed a split-Gal4 strategy to genetically intersect *dsx* expression with cell-type-specific transcription factors (Figures 2G, 2H, and S4A–S4C).

This approach successfully labeled all six *dsx*+ cell types and yielded cell counts consistent with prior anatomical estimates (Figure S4D).[10,44] Thus, every cell of a given type is represented in these intersections, unifying their transcriptional identity with anatomical identity and providing a powerful and unprecedented tool kit for future functional studies investigating *dsx*+ neuronal populations. A simultaneous publication independently identified similar findings in *D. melanogaster* and three other *Drosophila* species, highlighting the evolutionarily conserved nature of these transcriptional identities.[45]

Next, we used single-cell regulatory network inference and clustering (SCENIC) to identify gene regulatory networks (GRNs) that specify each *dsx*+ cell type (Figures 2I–2M). GRN-based subclustering paralleled transcriptional-based clustering (Figure 2J), underscoring the robustness of regulon architecture in defining neuronal identity. Many cell-type-enriched regulons corresponded to *dsx*+ cell-type-specific transcription factors (Figures 2F and 2K). We further examined the *otp* regulon, which was selectively enriched in pC1 neurons. Genes within this regulon were among the most specific pC1 markers (Figure 2L), implicating Otp as a core determinant of this key social arousal hub's identity, connectivity, and function.[46] Immunohistochemical analysis confirmed Otp expression in female and male pC1 neurons (Figures 2M and 2N), consistent with its predicted regulatory role. Together, these findings suggest that sex-specific circuit assembly is guided by Dsx and lineage-specific transcription factors such as Otp that shape neuronal subtype identity and connectivity.

### *fru*+ neurons developmentally diverge between the sexes

To investigate the role of *fru* in sex differences in central brain neurons, we generated a *fru*+ central brain neuron atlas by subclustering all *fru*-expressing cells (Figures 3A, 3B, and S5). Kenyon cells and monoaminergic neurons were analyzed separately (Figure S6). The cellular representation of *fru* in our atlas falls between previous estimates derived from antibody-based methods and *fru*$^{GAL4}$ reporter approaches.[8,47] In our companion paper,[19] we propose that the anatomical identity of transcriptionally defined cell types in our atlas, by and large, represents developmental hemilineages. Based on this, we estimate that *fru* is expressed within at least 57% of central brain hemilineages, consistent with its role as a temporal transcription factor marking subpopulations that reflect birth order.[19]

Using established markers, we annotated known neuronal populations, such as motor, neuroendocrine, and clock neurons

---

(F) Dot plots of the expression of significantly enriched transcription factors (left) and transmembrane receptors (right) across *dsx*+ neuronal cell types.

(G) Violin plots showing expression of key transcription factors, labeling distinct *dsx*+ neuronal cell types.

(H) Whole-mount immunofluorescence showing neuronal populations in the central brain identified via genetic intersection between *dsx* and key transcription factors. Images have been segmented; full expression patterns are shown in Figure S4C.

(I) Network diagram illustrating putative regulatory relationships between GRNs identified across *dsx*+ neurons. Nodes are colored based on GRN expression across *dsx*+ subtypes.

(J) GRN-tSNE of *dsx*+ neuronal cell type, colored by GRN-defined cell types.

(K) Dot plots of significantly enriched regulons across sex-merged *dsx*+ neuronal cell types (left) and between the sexes (right).

(L) *otp* GRN identified in *dsx*+ pC1 neurons.

(M) UMAPs showing the expression of *otp* in female (left) and male (right) *dsx*+ neurons.

(N) Immunofluorescence showing *dsx*+ pC1 neurons (GFP, left), Otp expression (blue, middle), and co-expression (merged, right). pC1 soma and hemilineage-associated axonal tracks were segmented from *dsx*$^{Gal4}$>mCD8::GFP.

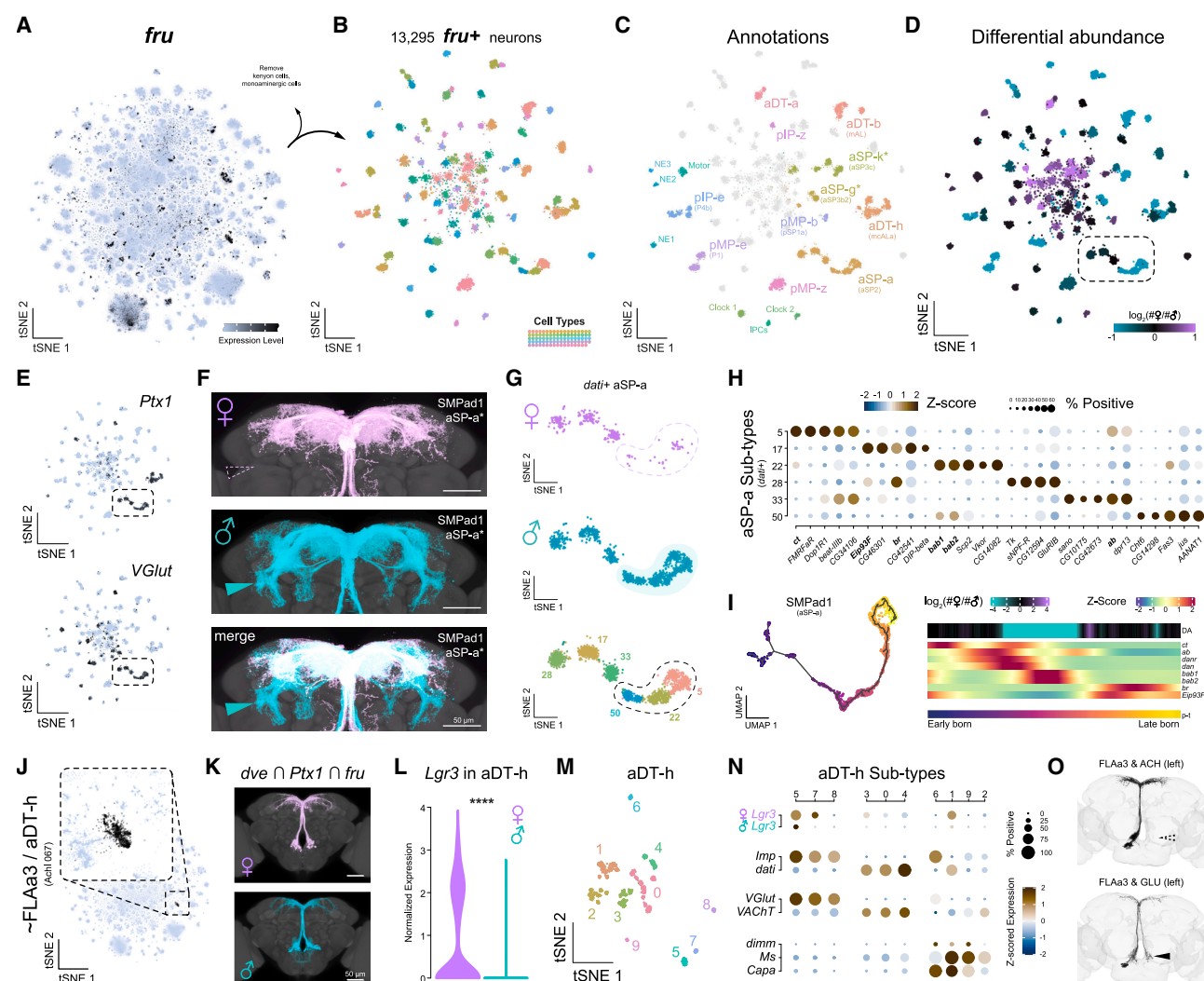

**Figure 3. fru+ meta-analysis reveals sexually diverse cell types in the adult central brain**

(A) tSNE of fru+ expression across the *Drosophila* central brain atlas.

(B) tSNE of 13,295 fru+ neurons subclustered and colored by transcriptionally unique cell type. Kenyon cells and monoaminergic cells were removed (see Figure S6).

(C) Annotated tSNE of anatomically defined fru+ neuronal cell types.

(D) tSNE of differential abundance in cell numbers between the sexes in fru+ cell types. Dashed box corresponds to box in (E).

(E) tSNEs of *Ptx1* and *VGlut* expression in fru+ neurons, with overlapping expression highlighted by dashed boxes.

(F) Immunofluorescence images of *Ptx1∩VGlut* neurons in female (magenta, top) and male (blue, middle) central brains, with merged overlay (white, bottom). Solid blue arrowhead indicates male-specific neurites, and dashed magenta arrowhead marks their absence in females (segmented; full patterns are shown in Figure S8).

(G) Zoomed-in region of the *dati*+ (late-born) aSP-a cell type within the fru+ central brain tSNE, showing female (top) and male (middle) cell distribution across transcriptionally distinct aSP-a subtypes (bottom). Male-specific subtypes (5, 22, and 50) are highlighted.

(H) Dot plot of significantly enriched gene expression in *dati*+ aSP-a subtypes. Temporal transcription factors (TFs) identified over pseudotime are bolded (see I).

(I) Pseudotime trajectory (left) of SMPad1 hemilineage (containing aSP-a neurons). Heatmap (right) of sex differences in cell abundance (top) and gene expression dynamics of transcriptional markers (middle) along pseudotime (p-t; bottom), showing key TFs defining early, middle, and late birth-order identities.

(J) Central brain tSNE highlighting fru+ aDT-h/FLA3 hemilineage, corresponding to Achl 067 cell type (black), with zoomed-in inset (dashed box).

(K) Immunofluorescence-based anatomical identification of aDT-h neurons using *dve*, *Ptx1*, and *fru* co-expression in female (top) and male (bottom) central brains (segmented; full patterns are shown in Figure S8).

(L) Violin plot of *Lgr3* expression in aDT-h neurons, showing significant female-bias (****$p < 0.0001$).

(M) UMAP subclustering of aDT-h neurons, colored by transcriptionally distinct subtypes.

(N) Dot plot of gene expression across aDT-h subtypes, highlighting subtype-specific expression of *Lgr3* (in females and males), early- vs. late-born markers (*Imp* and *dati*), neurotransmitter marker genes (*VGlut* and *VAChT*), and neurosecretory-specific TF *dimm* and neuropeptides (*Ms* and *Capa*).

(O) Electron microscopy (EM) reconstructions of FLAa3 neurons in one hemisphere of the central brain (FlyWire), showing distinct cholinergic (ACH) and glutamatergic (GLU) populations, highlighting anatomically distinct neurites (arrowheads).

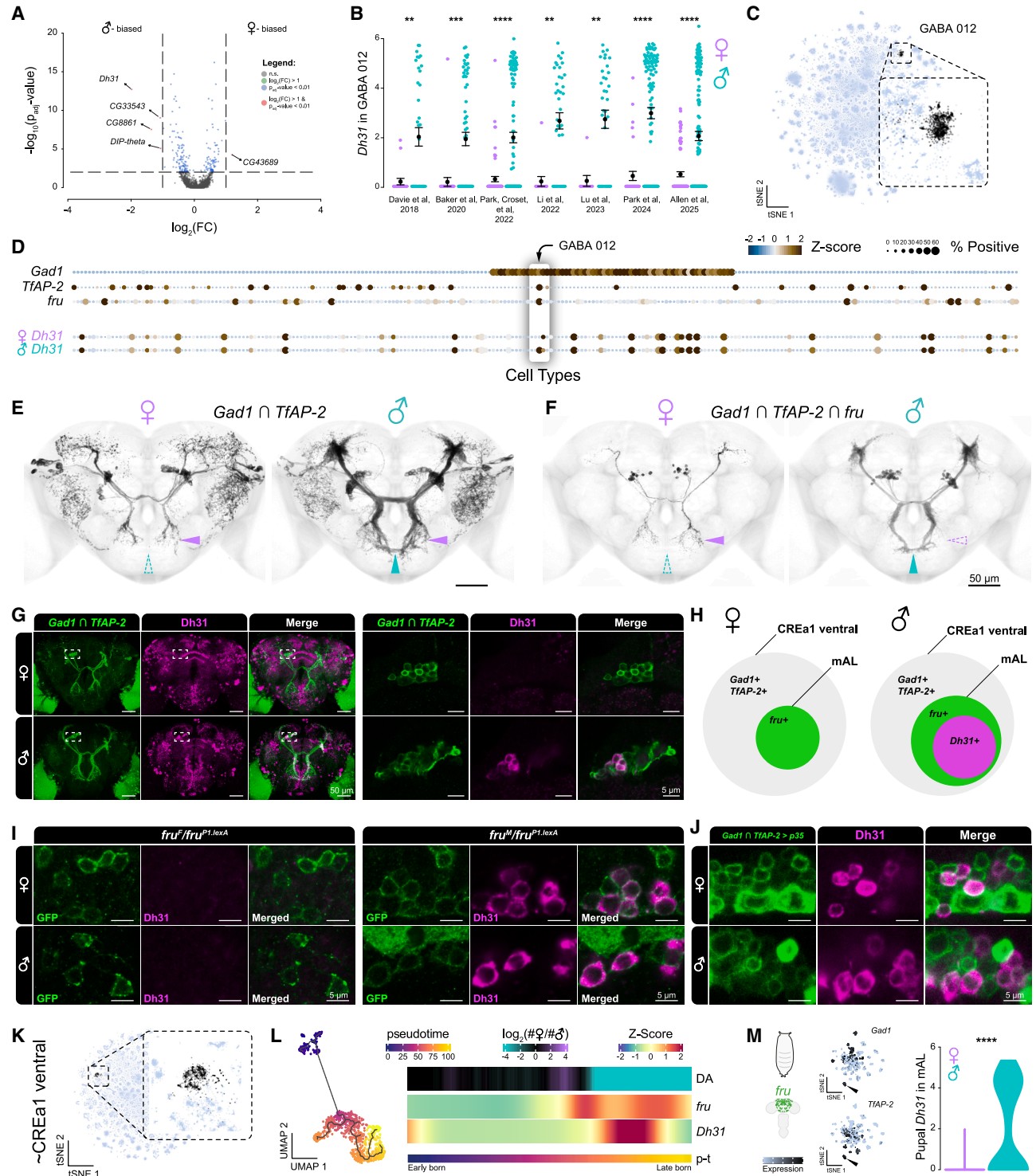

**Figure 4. Male-specific expression of the neuropeptide Dh31 in the central brain**

(A) Volcano plot of differentially expressed genes exhibiting male-biased (left) and female-biased (right) expression within the GABAergic 012.

(B) *Dh31* expression between females and males within the GABA 012 cluster, split by datasets. Colored points respresent individual cells; black points represent mean ± SEM (**p < 0.01, ***p < 0.001, and ****p < 0.0001, Bonferroni-corrected Wilcoxon signed-rank test).

(C) tSNE highlighting GABA 012 (in black) within the neuronal central brain atlas, with zoomed-in inset (dashed box).

(D) Dot plot of GABA 012-defining genes (*Gad1*, *TfAP-2*, and *fru*; top) and *Dh31* (split by sex; bottom) across neuronal clusters in the central brain.

*(legend continued on next page)*

(Figures 3C and S5; Table S2). Other transcriptionally defined clusters were annotated through genetic intersectional strategies, linking them to anatomically characterized *fru*+ neurons (Figure S7; Table S2). This approach provided a systematic pipeline for linking transcriptional profiles with anatomical *fru*+ populations. Notably, it identified both well-characterized (e.g., aSP-a and aDT-h) and previously unannotated *fru*+ cell types (pMP-z and pIP-z), highlighting its utility in uncovering the full diversity of sexually dimorphic neuronal populations.

Several *fru*+ neuronal cell types exhibited sex bias in cell numbers (Figure 3D). We focused on a prominent male-biased population corresponding to a central brain cell type with one of the highest numbers of DEGs (Figure 1H, cluster Glut 058). Using a double (*Ptx1* and *VGlut*) and triple (*fru*, *Ptx1*, and *VGlut*) genetic intersection strategy, we identified the glutamatergic *fru*+ aSP-a cell type (also known as aSP2; Figures 3E–3G, S8A, S8D, and S8E), one of the largest *fru*+ populations in the central brain.[48] Indeed, most subtypes of the SMPad1 hemilineage correspond to *fru*+ aSP-a neurons (Figure S8A). These neurons are localized in the superior medial protocerebrum (SMP) and form male-specific, characteristic ring-shaped arborizations in the lateral protocerebral complex (Figure 3F), a region densely innervated by *dsx*+ neurons (Figure 2H).[49,50]

Within the aSP-a clusters, we identified six transcriptionally distinct subtypes, three of which are male specific and likely contribute to the formation of male-specific arborizations (Figures 3E–3H). Notably, these subtypes express *abrupt* and *beat-III* genes, which are known to harbor sex-biased enhancer elements that exhibit differential chromatin accessibility in aSP-a neurons.[26] Our companion paper[19] shows that neuronal subtypes within a hemilineage are defined by temporal transcription factors associated with birth order that are retained in the adult. Building on this framework, we analyzed the entire SMPad1 hemilineage across pseudotime (Figure 3I), which orders cells along a continuous trajectory based on gene expression profiles. Male-specific subtypes emerged during a developmental window coinciding with the expression of *bab1* and *bab2*, temporal transcription factors that mark mid-to-late neurogenesis. This suggests that female neurons born during this window are eliminated via programmed cell death (PCD), a process that Fru is known to inhibit in males.[27,35]

Next, we focused on an entirely *fru*+ central brain cell type found to have dramatically sex-biased gene expression (Figures 1H, 3J, S1J, and S8B). Genetic intersection of transcription factors co-expressed in this cell type, *Ptx1* and *dve*, identified the previously described *fru*+ aDT-h (also known as mcALa[51]) neuronal population, corresponding to hemilineage FLAa3 (Figures 3K, S8B, and S7F). While this population is largely monomorphic,[48] we and others[52,53] observed a slight male bias in cell number alongside distinct neurites in the gnathal ganglia that form male-specific, horsetail-like dendritic fields (Figure 3K). As expected, the *Insulin-like peptide 8* (*Ilp8*) receptor *Lgr3*, a Fru target previously shown to be repressed in aDT-h neurons,[54] exhibited strong female-biased expression (Figures 3L and S8C).

Subclustering of aDT-h neurons revealed nine subtypes with distinct transcriptional signatures (Figure 3M). Unexpectedly, we observed female-specific *Lgr3* expression across three distinct subtypes: an early-born glutamatergic, a late-born cholinergic, and an endocrine subtype (Figure 3N). Exploring the FLAa3 hemilineage in the female connectome[55,56] revealed two anatomically distinct subclasses corresponding to differential neurotransmitter usage, which reshapes our understanding of information flow within this hemilineage: a cholinergic subtype whose projections terminate in the dorsal-most part of the SMP and a glutamatergic subtype that extends projections to the contralateral flange region, via the dorsal SMP, forming output sites in both neuropil regions (Figure 3O). Notably, the *Lgr3*+ endocrine subtype co-expresses the neuropeptides *Myosuppressin* (*Ms*) and *Capability* (*Capa*) in both sexes (Figure 3N). These findings suggest that *Ilp8* signaling may modulate the release of multiple neuropeptides in a female-specific manner, influencing the physiological state of females.[57,58]

### Male-specific neuropeptide expression via Fru-dependent neuronal survival

While most central brain neuronal cell types displayed few or no DEGs between the sexes (Figure 1H; Table S1), specific cell types exhibited dramatic sex differences in the expression of individual genes. A novel finding is the highly male-biased expression of the neuropeptide *Diuretic hormone 31* (*Dh31*) in a single GABAergic neuronal population (GABA 012) (Figures 4A–4C).

(E) Immunofluorescence of *TfAP-2*∩*Gad1* intersection identifies the CREa1 ventral hemilineage. Arrowheads highlight anatomical sex differences in neurite projections: magenta, female; blue, male; solid, present; dashed, absent (segmented; full patterns are shown in Figure S9A).

(F) Triple *Gad1*∩*TfAP-2*∩*fru* intersection identifies the sexually dimorphic mAL neuronal cluster (segmented; full patterns are shown in Figure S9A).

(G) Whole-mount immunofluorescence images of *Gad1*∩*TfAP-2*>myr::GFP (green), anti-Dh31 (magenta), and merged (white). Female (top), male (bottom), whole brains (left), and white dashed box insets highlighting individual section soma (right).

(H) Venn diagram of the CREa1 ventral hemilineage (gray), which encompasses *fru*+ mAL cells (green), with a subset of male mAL neurons expressing Dh31 (magenta).

(I) Dh31 expression in the mAL in hemizygous *fru*F females and males (*fru*F/*fru*P1.LexA; left) and in hemizygous *fru*M females and males (*fru*M/*fru*P1.LexA; right). Close-up of individual sections of mAL soma (green), anti-Dh31 (magenta), and merged. Female (top) and male (bottom). Whole brains are shown in Figure S9C.

(J) Immunofluorescence of central brains with *Gad1*∩*TfAP-2* intersection expressing p35 and myr::GFP (green), anti-Dh31 (magenta), and merged. Female (top) and male (bottom). Whole brains are shown in Figure S9E.

(K) Central brain neuron tSNE highlighting the mAL containing CREa1 ventral hemilineage (black), with zoomed-in inset (dashed box).

(L) Pseudotime trajectory (left) of CREa1 ventral hemilineage. Heatmap (right) showing sex differences in cell abundance (top) and gene expression dynamics of *fru* and *Dh31* (middle) along pseudotime (p-t; bottom).

(M) Schematic of mid-pupal, central brain, *fru* neurons (left), with tSNEs (middle) highlighting *Gad1* (top) and *TfAP-2* (bottom) expression. Arrowhead highlights mAL cell cluster. Violin plot of *Dh31* expression across sexes in the pupal mAL (right). ****p < 0.0001, Wilcoxon signed-rank test. Reprocessed data from Palmateer et al.[31] (see Figure S10).

Male-biased expression was consistently observed across all independent scRNA-seq datasets within our atlas (Figure 4B), confirming its robustness and biological relevance. Dh31 is critical in regulating various physiological processes in *Drosophila* and is homologous to the vertebrate calcitonin gene-related peptide (CGRP).[59]

To determine the anatomical identity of this GABAergic population, we identified the GABAergic marker *Gad1*, the transcription factors *TfAP-2*, and *fru* as robust markers of its transcriptional identity (Figure 4D). The genetic intersection of *TfAP-2* and *Gad1*, along with a triple intersection with *fru*, anatomically mapped this transcriptionally distinct cell type to the CREa1 ventral hemilineage (aka CREa1B) and its *fru*+ subset mAL (Figures 4E, 4F, and S9A). mAL neurons represent a well-characterized sexually dimorphic population, known for their role in processing and integrating sex-pheromone cues in males, thereby ensuring that courtship behavior is directed toward sexually receptive, species-appropriate mates.[60–63]

To validate the sex-specific expression of Dh31, we performed Dh31 immunostaining, which revealed male-specific expression in CREa1 ventral neurons, specifically within the subset of *fru*+ mAL neurons (Figures 4G, 4H, and S9B). To assess whether this sex difference is directly regulated by *fru*, we utilized *fru* splice mutants that induce constitutive splicing to either female (non-functional, *fru*F) or male (functional, *fru*M) isoforms.[64] Chromosomal females expressing the functional male isoform of *fru* (*fru*M/*fru*P1.lexA) exhibited Dh31 expression in mAL neurons, whereas chromosomal males expressing the non-functional female isoform of *fru* (*fru*F/*fru*P1.lexA) did not (Figures 4I, S9C, and S9D). These findings demonstrate that male-specific Dh31 expression depends on the functional male isoform of Fru (Fru M).

Fru protects male mAL neurons from PCD, leading to fewer mAL neurons in females.[35] To investigate whether PCD influences *Dh31* expression in females, we inhibited cell death using the p35 apoptosis inhibitor, which restored Dh31 expression in female mAL neurons (Figures 4J and S9E). This result indicates that the absence of Dh31 in female mAL neurons is due to female-specific PCD rather than male-specific *fru*-mediated transcriptional regulation of *Dh31* and is supported by simultaneous findings in another study.[65] Pseudotime analysis suggests that *fru* and *Dh31* co-positive, male-specific cells are born late in the CREa1 ventral hemilineage (Figures 4K and 4L). Furthermore, reanalysis of a pupal *fru*+ scRNA-seq dataset[31] (Figure S10) revealed that mAL neurons exhibit male-biased *Dh31* expression during mid-pupal development (Figure 4M), suggesting that female-specific cell death of *Dh31*+ neurons occurs before 48 h post-puparium formation. To explore potential downstream targets of local Dh31 release, we genetically intersected its receptor, *Dh31-R*, in both *dsx*+ and *fru*+ neurons, revealing broad expression in males. Thus, Dh31 signaling likely coordinates sex-specific neuronal activity across multiple brain circuit elements (Figure S9F).

### Interplay between Dsx and Fru defines sexually dimorphic neuronal lineages

To investigate how *dsx* and *fru* collectively orchestrate the specification of sexually dimorphic neuronal identities, we merged their respective subatlases to construct a unified *dsx*+ and *fru*+

central brain atlas, encompassing all transcriptionally dimorphic neurons (Figures 5A and 5B). We then isolated cells within defined hemilineages—the developmental units comprising the central brain—that co-express *dsx* and *fru* to systematically explore their transcriptional relationships. Leveraging genetic intersectional tools, prior MARCM clonal analyses, and hemilineage-associated tract annotations from the connectome, we annotated and confirmed six hemilineages that co-express *dsx* and *fru* (Figures 5B–5D and S11).[27,44,48,51,53,55,56,66–68]

Within each of these hemilineages, we resolved distinct neuronal subtypes co-expressing *dsx* and *fru* in addition to subtypes expressing each transcription factor independently (Figures 5E and S12). This finding indicates that *dsx* and *fru* can act cooperatively and independently within hemilineages to diversify neuronal identities. In our companion study,[19] we found that transcriptional profiles in the adult brain retain signatures of developmental timing, enabling systematic reconstruction of neuronal birth order across hemilineages (Figures 5F and 5G). This developmental structure provides a framework for understanding how sex-determining transcription factors interact with temporal patterning mechanisms. To investigate the inferred developmental dynamics between *dsx* and *fru*, we extracted and performed pseudotime analysis on these six hemilineages (see STAR Methods), revealing striking lineage-specific variation in the onset and overlap of their expression (Figure 5H). The lack of chronological coordination of *dsx* and *fru* between hemilineages suggests that the precise mechanisms by which Dsx and Fru act in concert to regulate neuronal identity and sex-specific differentiation during neurogenesis occur in a hemilineage-specific manner. Future experiments into these developmental dynamics will elucidate these patterns.

### Developmental timing and PCD shape sex differences in pC1/P1 neurons

We observed considerable transcriptional heterogeneity among dimorphic neurons within each *dsx*+/*fru*+ hemilineage. To explore this diversity, we focused on the pC1/P1 cluster within the DM4 dorsal hemilineage, a critical integrative hub for social arousal in both sexes (Figure 6A). This cluster comprises anatomically and functionally distinct neuronal subtypes that differ in abundance and connectivity between the sexes, properties previously shown to depend on *dsx* and *fru*.[10,30,49,69–76] We identified nine broad transcriptionally distinct pC1/P1 subtypes (Figures 6B and 6C) and confirmed they correlated with morphologically distinct neuronal populations by genetic intersectional mapping (Figures 6D and S13). This reinforces the finding that pC1/P1 subtypes are functionally specialized; indeed, a more refined analysis of this population reveals extensive transcriptional and subtype diversity, explored in detail in Figure S12.

When analyzing DEGs between the sexes across the central brain, we found that the cell type displaying the highest number of sex-biased genes as well as the most dramatic differential cell abundance between the sexes corresponds to pC1/P1 neurons (Achl 058; Figure 1H). Several of these DEGs encode temporal transcription factors that regulate neuron birth order and subtype identity during neurogenesis (Figure 6E). To explore the inferred developmental dynamics within the pC1/P1 neuronal population, we conducted a pseudotime analysis, revealing previously

**CellPress** OPEN ACCESS

**Cell Genomics**
**Article**

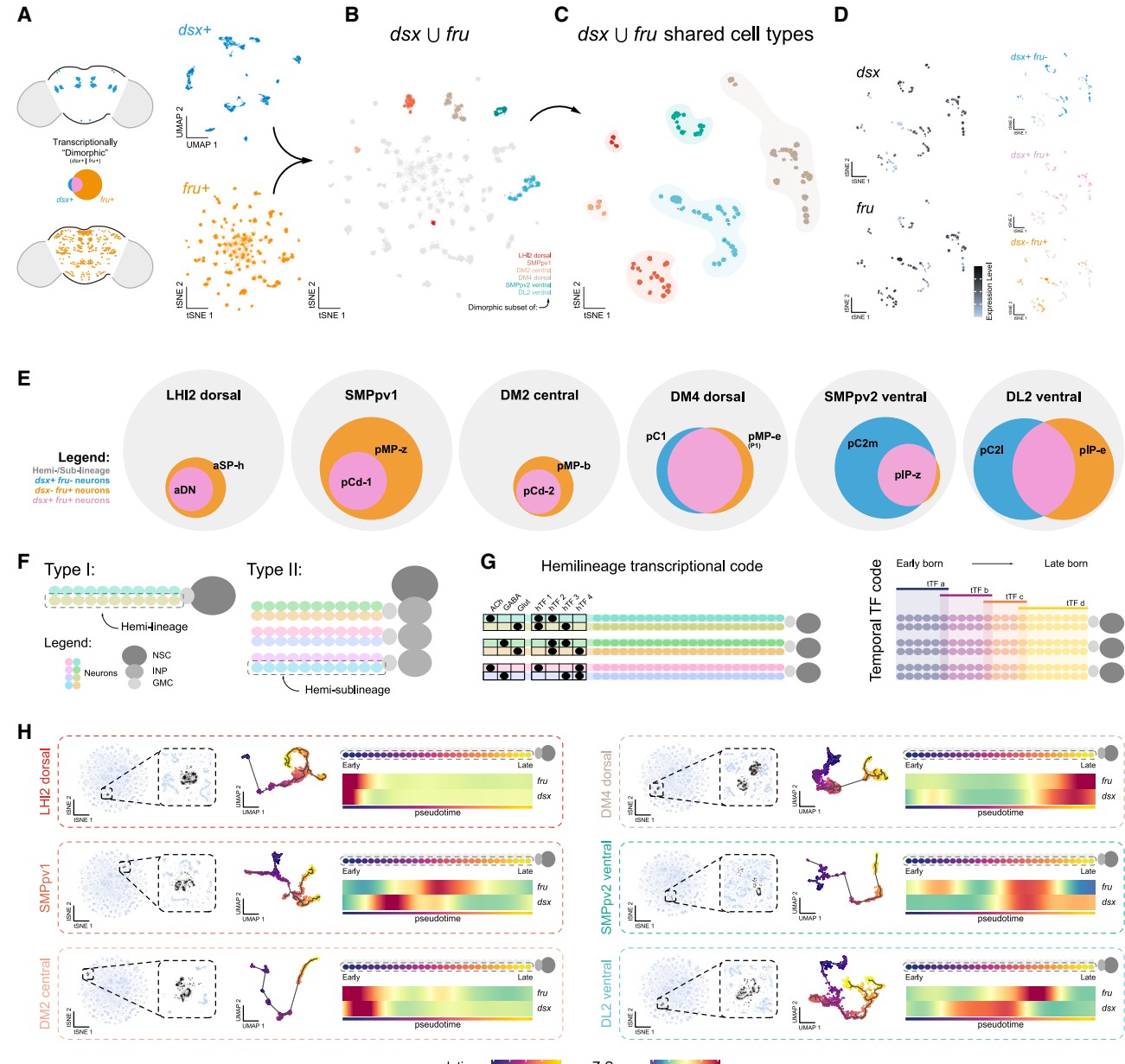

**Figure 5. Developmental relationship between *dsx* and *fru* cell types in the central brain**

(A) Schematic and neuronal subatlases of *dsx*⁺ (blue, top) and *fru*⁺ (orange, bottom) within the central brain.

(B) tSNE of *dsx*⁺ and *fru*⁺ union central brain atlas (*dsx*U*fru*), colored by cell types with shared hemilineage identities (LHl2 dorsal, SMPpv1, DM2 central, DM4 dorsal, SMPpv2 ventral, and DL2 ventral).

(C) tSNE of subclustered *dsx*U*fru* neurons within shared hemilineage cell types (indicated by colors).

(D) tSNE visualization of *dsx* and *fru* expression within shared hemilineage cell types (left), with individual and co-expressed cells shown in different colors (right).

(E) Venn diagrams depicting the relationship between *dsx*⁺ and *fru*⁺ expressing neurons within distinct hemilineages.

(F) Schematic of *Drosophila* hemilineage neurogenesis from type I and type II neural stem cells (NSCs) (GMC, ganglion mother cell; INP, intermediate neural progenitor).

(G) Combining neurotransmitter identity markers with hemilineage-restricted (hTFs) and temporally restricted transcription factors (tTFs) provides a framework for identifying transcriptionally distinct neuronal subtypes across the central brain.

(H) Pseudotime analyses of six *dsx*⁺/*fru*⁺-containing hemilineages in the central brain. tSNEs highlighting hemilineages in reprocessed central brain atlas (left, see STAR Methods). Pseudotime trajectories (middle) of hemilineages and heatmaps (right) showing gene expression dynamics of *dsx* and *fru* along pseudotime (from early- to late-born neurons).

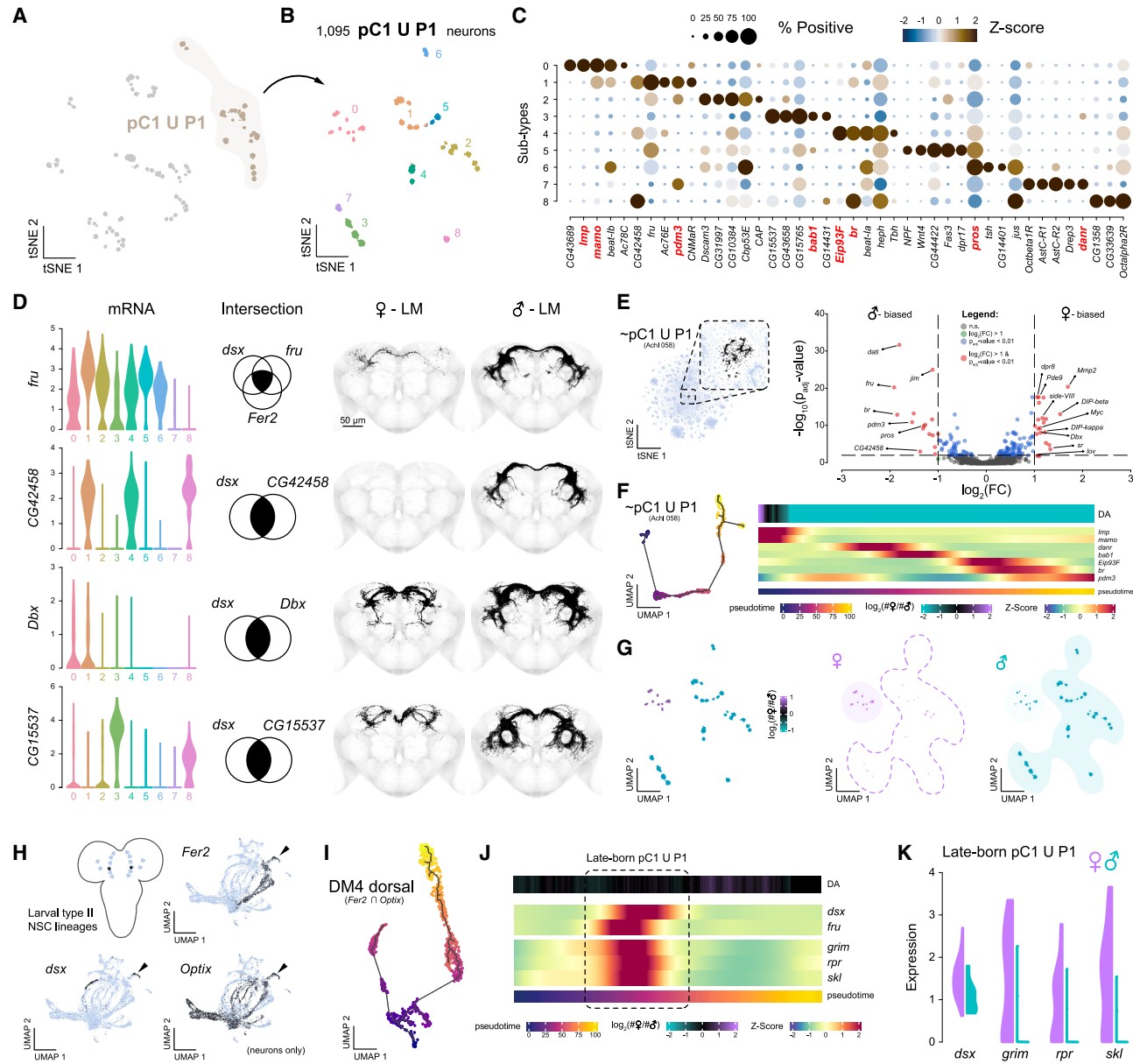

**Figure 6. Genetically defining pC1/P1 neurons uncovers sex-specific cell death**

(A) tSNE of *dsx*∪*fru* neurons with shared hemilineage identities, with the pC1/P1 subtypes highlighted.

(B) UMAP of pC1/P1 neuron subclustering, colored by transcriptionally distinct cluster identities.

(C) Dot plot of top genes defining pC1/P1 subtypes. Developmental genes predicted to mark birth order within hemilineages[19] are highlighted in red.

(D) Violin plots of pC1/P1 subtype-defining gene expression (left). Genetic intersections identifying anatomically distinct cell types (middle). Immunofluorescence light microscopy (LM) images showing pC1/P1 neuronal populations identified via genetic intersection (right) in females and males (segmented; full patterns are shown in Figure S13).

(E) tSNE (left) highlighting the pC1/P1-containing cell type AchI 058 in the neuronal central brain atlas with zoomed-in inset (dashed box). Volcano plot (right) showing differentially expressed genes in the AchI 058 cluster between sexes.

(F) Pseudotime trajectory (left) of AchI 058 cell type. Heatmaps (right) showing sex differences in cell abundance (top) and gene expression dynamics of key genes defining early, middle, and late birth-order identities (middle) along pseudotime (bottom).

(G) UMAPs of differential abundance between the sexes in pC1 subtypes (left) and split by sex (right). Male-specific subtypes are highlighted within the dashed line.

(H) Schematic of larval type II neuroblast lineages (top left). UMAPs of *Fer2*, *Optix*, and *dsx* expression in type II lineages (right, bottom) defining the DM4 dorsal hemilineage (arrowhead). Reprocessed data from Michki et al.[77] and Rajan et al.[78]

*(legend continued on next page)*

established temporal transcription factors[19] expressed in distinct pseudotime windows (Figure 6F). Comparing developmental trajectories with differential cell abundance between sexes, we found a female-biased temporal window that correlated with markers of early-born neurons (*Imp* and *mamo*), as well as a male-specific temporal window that correlated with mid- and late-born markers (Figure 6F). When examining the sex distribution among pC1/P1 subtypes, we found that all but one subtype was male specific (Figure 6G). Thus, female and male pC1/P1 neurons arise from distinct temporal windows and represent non-overlapping neuronal subtypes.

To further resolve how developmental timing intersects with sexual identity, we analyzed larval scRNA-seq datasets from type II neuroblasts and their lineages, which include the DM4 dorsal hemilineage that gives rise to pC1/P1 neurons.[27,77,78] We extracted DM4 dorsal cells and performed a pseudotime analysis, identifying a developmental window during which *dsx* and *fru* expression peaks (Figures 6H–6J). While examining genes that co-varied with *dsx* and *fru*, we identified three key pro-apoptotic genes: *grim*, *rpr*, and *skl* (Figure 6J). Notably, the expression of these genes was female specific in these post-embryonic, late-born pC1/P1 neurons (Figure 6K). Previous work demonstrated that both sexes experience PCD within the pC1/P1 cluster, with females experiencing more extensive neuronal loss than males, leading to highly sexually dimorphic numbers of neurons[27] (Figure S4D). Our data refine this model, suggesting that female-specific PCD primarily eliminates late-born pC1/P1 neurons, whereas male-specific PCD targets early-born pC1/P1 neurons. Furthermore, we detected no sex difference in DM4 dorsal neuron abundance at late larval stages (Figure 6J), indicating that sex-biased PCD initiates during late larval development but completes post-puparium formation. Together, these data support a model in which birth order within a hemilineage dictates sexual fate via temporally gated, sex-specific PCD.

### Sexually dimorphic cells represent distinct subpopulations between the sexes

In our companion paper, we show that neurons born at different developmental time points within a hemilineage (Figures 5F and 5G), early vs. late, acquire distinct transcriptional signatures that define their mature identities.[19] We further demonstrate that persistent expression of *Imp* and *dati* reliably marks early- and late-born neurons, respectively, providing effective proxies for temporal identity. To examine the broader implications of birth order and sex-specific differentiation in the central brain, we first analyzed differential cell abundance across *dsx*+ and *fru*+ populations (Figure 7A). We found that female-biased neuronal types predominantly expressed the early temporal marker *Imp*, whereas male-biased types were enriched for the late-born marker *dati* (Figure 7B). Indeed, we observed a strong correlation between sex-biased cell numbers and these birth-order markers (Figure 7C), suggesting temporal identity is broadly linked to sexual dimorphism across neuronal populations.

To investigate this relationship across the entire central brain, we visualized sex differences in cell abundance, sexually dimorphic (*dsx*+ or *fru*+) vs. monomorphic (*dsx*− and *fru*−) cell types, and *Imp* and *dati* expression (Figures 7D–7F). Once again, we found a consistent correlation between sex-biased neuronal populations and birth order (Figure 7G). Early-born neurons were overrepresented among female sexually dimorphic neurons compared to male dimorphic neurons, while no sex difference in early vs. late markers was observed among monomorphic populations. Comparing within-sex and across sexually dimorphic vs. monomorphic neurons revealed that female dimorphic neurons have an overrepresentation of early-born neurons relative to monomorphic female neurons. Conversely, male dimorphic neurons have an underrepresentation of early-born neurons compared to male monomorphic neurons.

These findings support a model where sexually dimorphic neurons arise through birth-order-dependent PCD within shared hemilineages (Figure 7H). In this framework, female- and male-specific neurons represent developmentally distinct subpopulations within a hemilineage, defined by sex-specific survival rather than by the generation of entirely distinct cell types. Thus, sexually dimorphic neurons, including the pC1 cluster, may be more accurately described as "paralogs" between the sexes rather than "orthologs" arising from common progenitors but diverging via sex-biased elimination. Given that birth order is second only to hemilineage as a determinant of neuronal identity in the central brain, small sex-specific modifications in neuronal survival within a hemilineage lead to profound differences in circuit architecture and behavior.

### DISCUSSION

It is widely assumed that female and male nervous systems are composed of equivalent underlying cell types, with sex-specific features arising from differences in wiring, abundance, or physiology. Our findings challenge this view. By analyzing sexed single-cell transcriptomes of the adult *Drosophila* central brain, we show that sexual dimorphism largely arises via the selective survival of distinct subpopulations of neurons within shared developmental lineages. By integrating transcriptional identity, sex, and markers of developmental histories, we provide a novel understanding of how behavioral diversity arises from a common developmental framework.

We took a data-driven approach to identify where sex influences neuronal identity across the adult central brain. Overall, sex-biased gene expression is limited, as only ∼1% of genes differ significantly between females and males, and most of these differences are modest in magnitude. However, when focusing on neuronal cell types showing the most pronounced differences in gene expression and cell abundance, we found that sexually dimorphic populations are consistently marked by expression of the sex-determination genes *dsx* and *fru*. Gratifyingly, the convergence between our unbiased transcriptomic

---

(I) Pseudotime analysis of DM4 dorsal hemilineage defined by *Fer2* and *Optix* co-expression.

(J) Heatmaps showing lack of sex differences in cell abundance (top) and gene expression dynamics of *dsx* and *fru* and cell death genes *grim*, *rpr*, and *skl* (middle) along pseudotime (bottom).

(K) Split violin plot of *dsx* and cell death gene expression in post-embryonic larval pC1/P1 neurons in females and males.

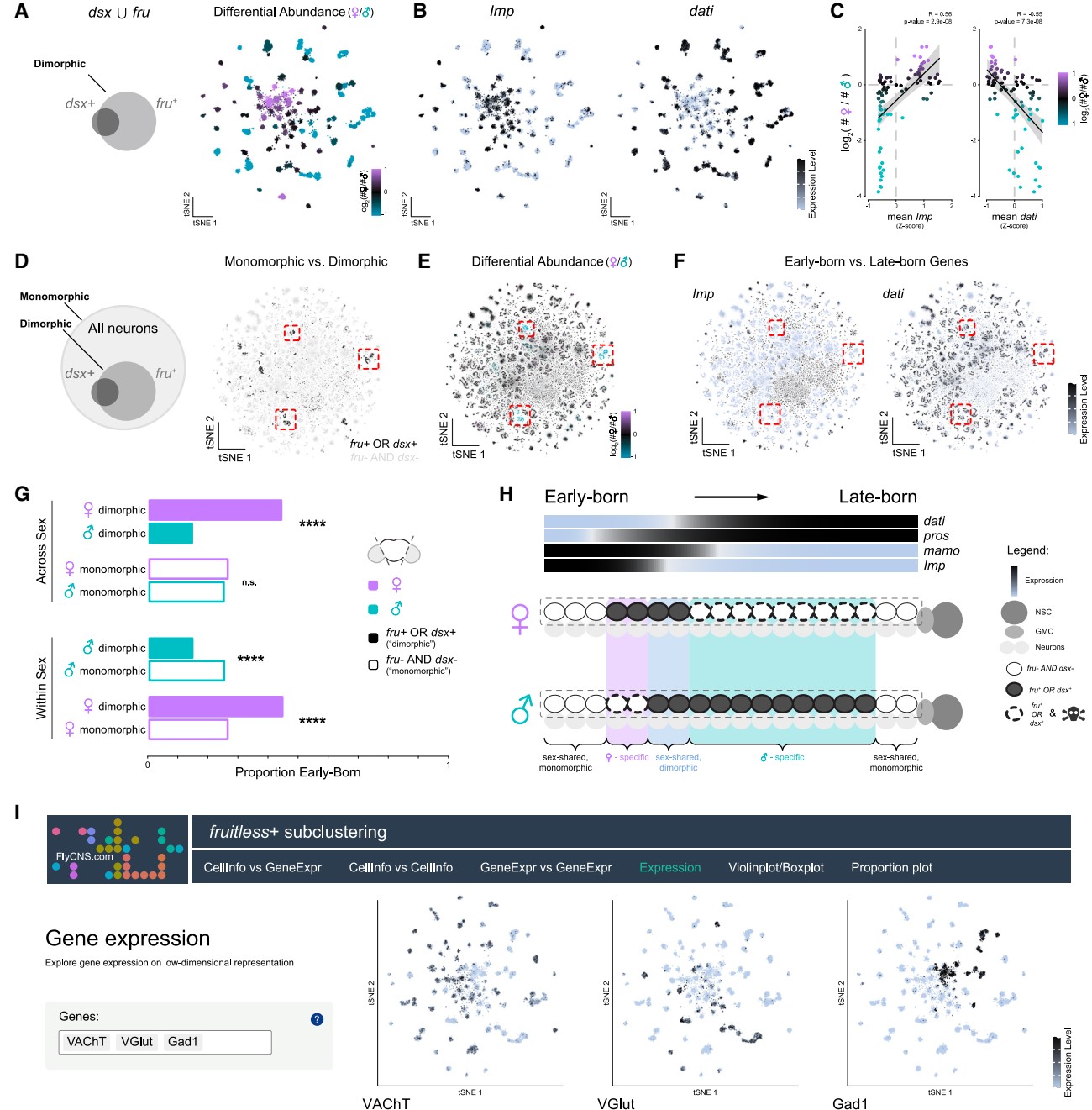

**Figure 7. Sexually dimorphic neurons experience differential cell death**

(A) Schematic (left) and tSNE (right) of *dsx∪fru* transcriptionally dimorphic neurons in the central brain, highlighting differential abundance in cell numbers between the sexes across cell types.

(B) tSNEs of *Imp* (left) and *dati* (right) expression across *dsx∪fru* neurons.

(C) Scatterplots showing correlations of sex-biased cell numbers with *Imp* (left) and *dati* (right) expression across *dsx∪fru* neuronal populations.

(D) Schematic (left) and tSNE (right) showing classification of neurons as either transcriptionally monomorphic or dimorphic based on *dsx* and *fru* expression in the central brain. Kenyon cells have been removed. Red dashed boxes highlight three regions.

(E) tSNE of differential abundance between the sexes across neuronal cell types in the central brain.

(F) tSNEs of early-born (*Imp*, left) and late-born (*dati*, right) marker gene expression across the central brain neurons, indicating developmental timing differences in neuronal populations.

(G) Proportion of early-born cells among dimorphic and monomorphic populations across sex (top) and within sex (bottom). See STAR Methods for details; ****$p < 0.0001$, n.s., not significant; Bonferroni-corrected chi-squared test.

*(legend continued on next page)*

analysis and historical forward genetic screens for sex-specific behaviors, which identified *dsx* and *fru*, validates both approaches and highlights the lasting insights offered by classical genetic approaches in behavioral neuroscience.[79,80]

Among the SDH genes, only *fru* and *dsx* show restricted, cell-type-specific expression across the central brain, while upstream regulators like *Sxl* and *tra* are ubiquitously expressed. This finding suggests that the nervous system is transcriptionally poised for sexual differentiation but that only select subsets of neurons implement this program via sex-specific transcriptional networks. Intriguingly, *dsx* expression is always accompanied by *fru* expression within the same hemilineage, with subsets of cells co-expressing both. Prior work has shown that Dsx and Fru may co-regulate downstream targets and even bind regulatory elements of other SDH genes, suggesting a combinatorial code that stabilizes sexual identity.[24,25] This modular control, consistent with the theory of facilitated variation,[81] enables small regulatory shifts in expression timing or pattern to yield large-scale effects on behavior.

Although the adult nervous system is generated through neurogenesis, it is sculpted by apoptosis, a mechanism used throughout the animal kingdom to generate alternative forms and functions.[82] Neurons born at different temporal windows face different survival outcomes in both sexes. We observe a striking sex bias in neuronal survival based on birth order, with female-biased neurons enriched among early-born subtypes and male-biased neurons among late-born ones (Figure 7). Rather than both sexes retaining and modifying equivalent neurons, our data show that females and males retain different subsets of neurons from the same hemilineage, shaped by sex-specific patterns of Dsx- or Fru-dependent cell death (Figure 6). Fru is known to protect neurons from apoptosis in males,[27,35] while Dsx has been shown to both inhibit and promote apoptosis, depending on the cellular context.[10,52,83–93] A novel finding in our study is the extent of Fru-dependent cell death, resulting in early-born neurons being retained in females but eliminated in males (Figure 7). Although Fru in males has been shown to induce PCD,[88] this remains an underappreciated mode of creating dimorphism. Future studies following the development of this circuitry will shed further light on when and where these mechanisms are differentially used within hemilineages.

The interplay between temporal identity and sex-specific factors shapes dimorphic outcomes, aligning with the principle that core developmental processes balance robustness and flexibility to allow neural circuits to adapt without compromising their fundamental structure.[94,95] In this framework, early-born, female-biased pathfinding neurons provide a stable scaffold, while late-born, male-biased neurons offer a flexible substrate for behavioral elaboration. This strategy of leveraging developmental pathways to selectively retain or prune temporal windows between the sexes may prove to be a model of exaptation,[96] whereby a trait adapted for one purpose, neurodevelopment,

is co-opted for another, sexual differentiation, and may facilitate the evolution of dimorphic behaviors.

Our findings recast sexual differentiation in the nervous system as a developmentally embedded, resource-efficient process. Rather than constructing *de novo* sex-specific architectures, the brain leverages conserved spatial and temporal logic, tuning outcomes via selective modulation of cell survival. This principle may generalize across species. In vertebrates, although hormonal cues dominate models of sexual differentiation, increasing evidence points to conserved developmental mechanisms, e.g., differential neurogenesis, maturation, PCD, and synaptogenesis, as key contributors to behavioral dimorphism.[97,98] As in *Drosophila*, these processes likely act on shared developmental templates, modified by sex at key regulatory nodes.

Our study acts as a powerful roadmap for future cross-species comparisons. We show lineage diversification without duplication; rather than evolving a new "female version" and "male version" of a circuit, a single hemilineage gives rise to both, with sex determining which neurons survive. This process is a resource-efficient evolutionary solution not limited to sexual differentiation and has also been noted in adaptive differences between species.[99] Implicit in this model is that neither the female nor the male brain represents the "default," but rather both are sculpted from a shared "template," resulting in two separate and distinct trajectories.

We have created a user-friendly website (https://www.flycns.com) to improve accessibility and usability for the wider community. This portal offers an interactive web-based visualization of the atlases referenced in this study (Figure 7I). We have shown that our atlas achieves the resolution necessary to detect subtle transcriptional differences in rare cell types by dramatically expanding the number of cells sampled from the adult brain. Importantly, we show that many sexually dimorphic neurons arise from different developmental windows within the same lineage and are not one-to-one correlates of one another. This finding suggests that sex-specific transcriptional differences often reflect divergent developmental histories, underscoring the need for future single-cell analyses encompassing a broad developmental window. As complete adult brain connectomes become available for both sexes,[56,100] our transcriptomic atlas will be a valuable resource enabling the linking of molecular and anatomical identities. Indeed, we consider transcriptomic identity to be an equally informative and distinct axis for defining cell types; thus, integrating both identities will facilitate a richer understanding of how gene expression and connectivity together shape sex-specific neural circuits and behavior.

### Limitations of the study

Most technical constraints mirror those in our companion study.[19] Integrating multiple, diverse datasets identified sex differences that are robust across genetic backgrounds and

---

(H) Proposed model of differential cell death in sexually dimorphic cells, with females experiencing more late-born cell death and males experiencing more early-born cell death. Top: expression dynamics of birth-order marker genes along the early-to-late neurogenesis axis. Bottom: schematic representation of hemilineage neurogenesis; cells with dashed borders represent sex-specific programmed cell death.

(I) Web application, available at flycns.com, provides an interactive interface for data exploration.

## Article

physiological states; however, it likely masked subtler sex-biased differences. Future studies will benefit from sample multiplexing to boost statistical power and reduce batch effects. As our adult analyses point to developmental origins of dimorphism, a sexed single-cell developmental atlas followed by *in vivo* functional studies will be necessary to determine the causal roles sex plays in neuronal cell-type specification in the central brain.

## RESOURCE AVAILABILITY

### Lead contact

Requests for further information, resources, and reagents should be directed to and will be fulfilled by the lead contact, Stephen F. Goodwin (stephen.goodwin@cncb.ox.ac.uk).

### Materials availability

All unique/stable reagents generated in this study are available from the lead contact without restriction.

### Data and code availability

Datasets described herein can be visualized at https://www.flycns.com/. Sequencing files, digital expression matrices, and Seurat RDS files are available from Gene Expression Omnibus (GEO; https://www.ncbi.nlm.nih.gov/geo/), accession number GSE296540. Code used in this analysis is available from GitHub (https://github.com/aaron-allen/Dmel-adult-central-brain-atlas) and Zenodo (https://doi.org/10.5281/zenodo.17513514). Light microscopy images are available at Virtual Fly Brain (https://v2.virtualflybrain.org). Previously published data used in this study are listed in the key resources table.

## ACKNOWLEDGMENTS

We thank E.J. Clowney, Y. Ding, F. Hirth, E. Rideout, J. Walsh, and members of the Goodwin lab for helpful discussions and critical reading of the manuscript. We are grateful to F. Casares, C. Desplan, J. Dubnau, T. Erclik, J. Hall, H. Lacin, G. Miesenböck, M. Nitabach, T. Shirangi, S. Waddell, U. Walldorf, and J. Wang for sharing stocks and reagents. Stocks obtained from Bloomington Drosophila Stock Center (NIH P40OD018537) were used in this study. A.M.A. and M.C.N. were supported by a Wellcome Trust Senior Investigator Award to S.F.G. (106189/Z/14/Z) and by a BBSRC grant (BB/X016595/1) to M.C.N., A.M.A., and S.F.G. T.N. and F.A. were supported by a BBSRC grant (BB/Y001869/1) to T.N. and S.F.G. A.M.A. was supported by a Wellcome Trust Collaborative Award to S.F.G. (209235/Z/17/Z).

## AUTHOR CONTRIBUTIONS

Conceptualization, A.M.A., M.C.N., and S.F.G.; methodology, A.M.A., M.C.N., T.N., F.A., and S.F.G.; investigation, A.M.A., M.C.N., T.N., and F.A.; resources, S.F.G.; writing, A.M.A., M.C.N., T.N., F.A., and S.F.G.; supervision, S.F.G.; funding acquisition, S.F.G.

## DECLARATION OF INTERESTS

The authors declare no competing interests.

## DECLARATION OF GENERATIVE AI AND AI-ASSISTED TECHNOLOGIES IN THE WRITING PROCESS

During the preparation of this work, the authors used ChatGPT, Google Gemini, and Grammarly to assist with code and grammatical text edits. The authors reviewed and edited the content as needed and take full responsibility for the publication's content.

## STAR★METHODS

Detailed methods are provided in the online version of this paper and include the following:

- ● KEY RESOURCES TABLE
- ● EXPERIMENTAL MODEL AND STUDY PARTICIPANT DETAILS
  - ○ Drosophila stocks
  - ○ Full genotype list
- ● METHOD DETAILS
  - ○ Generation of meta-central brain neuronal atlas
  - ○ Annotation of cell types and estimate of depth of coverage
  - ○ Annotation of sex
  - ○ Differential gene expression analysis
  - ○ Gene ontology analysis
  - ○ Subclustering *doublesex* and *fruitless* neurons
  - ○ Early- vs. late-born annotation
  - ○ Gene regulatory network analysis with SCENIC
  - ○ Pseudotime analysis
  - ○ Re-clustering and correcting for early- vs. late-born neurons
  - ○ Generating a web app
  - ○ Generation of split-Gal4 driver lines
  - ○ Immunohistochemistry
  - ○ Confocal image acquisition and processing

## SUPPLEMENTAL INFORMATION

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

## STAR★METHODS

### KEY RESOURCES TABLE

| REAGENT or RESOURCE | SOURCE | IDENTIFIER |
|---|---|---|
| **Antibodies** | | |
| anti-chicken Alexa Fluor 488 | Thermo Fisher Scientific | Cat# A-11039, RRID:AB_2534096 |
| anti-rabbit Alexa Fluor 488 | Thermo Fisher Scientific | Cat# A-11034, RRID:AB_2576217 |
| anti-mouse Alexa Fluor 546 | Thermo Fisher Scientific | Cat# A-11030, RRID:AB_2534089 |
| anti-rabbit Alexa Fluor 633 | Thermo Fisher Scientific | Cat# A-21070, RRID:AB_2535731 |
| anti-guinea pig Alexa Fluor 633 | Thermo Fisher Scientific | Cat# A-21105 RRID:AB_2535757 |
| anti-rat Alexa Fluor 633 | Thermo Fisher Scientific | Cat# A-21094, RRID:AB_2535749 |
| anti-GFP polyclonal (rabbit) | Thermo Fisher Scientific | Cat# A-6455, RRID:AB_221570 |
| anti-Brp (nc82) monoclonal (mouse) | Developmental Studies Hybridoma Bank | Cat# nc82, RRID:AB_2314866 |
| anti-GFP (chicken) | Abcam, UK | Cat# ab92456, RRID:AB_10561923 |
| anti-RFP (rabbit) | antibodies-online | Cat# ABIN129578, RRID:AB_10781500 |
| anti-FLAG (rat) | Novus Biologicals | Cat# NBP1-06712, RRID:AB_1625981 |
| anti-Otp polyclonal (guinea pig) | Gift from U. Walldorf | Hildebrandt et al., 2020[101] |
| anti-Dh31 polyclonal (rabbit) | Gift from J. Wang | Lin et al., 2022[102] |
| **Chemicals, peptides, and recombinant proteins** | | |
| Formaldehyde | Sigma-Aldrich | Cat# 47608-250ML-F |
| Phosphate buffered saline (PBS) | Sigma-Aldrich | Cat# P3183-10PAK |
| Triton X-100 | Sigma-Aldrich | Cat# T8787-100ML |
| Normal Goat Serum | Sigma-Aldrich | Cat# G9023 |
| Vectashield mounting medium | Vector Laboratories | Cat# H-1000, RRID:AB_2336789 |
| **Deposited data** | | |
| Central brain scRNA-seq | Allen et al., 2025a[19] | GEO: GSE296540 |
| Whole brain scRNA-seq | Davie et al., 2018[103] | GEO: GSE107451 |
| Whole brain scRNA-seq | Baker et al., 2021[104] | GEO: GSE152495 |
| Central brain scRNA-seq | Park et al., 2022[105] | GEO: GSE207799 |
| Central brain scRNA-seq | Dopp et al., 2024[106] | GEO: GSE221239 |
| Central brain snRNA-seq | Lee et al., 2025[107] | GEO: GSE247965 |
| Fly Cell Atlas, head snRNA-seq | Li et al., 2022[18] | ENA: E-MTAB-10519 |
| Aging Fly Cell Atlas, head snRNA-seq | Lu et al., 2023[108] | GEO: GSE218661 |
| Alzheimer's Fly Cell Atlas, head snRNA-seq | Park et al., 2025[109] | GEO: GSE261656 |
| Larval type II lineages | Michki et al., 2021[77] | GEO: GSE153723 |
| Larval type II lineages | Rajan et al., 2023[78] | GEO: GSE218257 |
| Pupal CNS scRNA-seq | Brovero et al., 2021[110] | GEO: GSE162098 |
| Optic lobe scRNA-seq | Ozel et al., 2021[111] | GEO: GSE142789 |
| Pupal FACS *fru+* scRNA-seq | Palmateer et al., 2023[31] | GEO: GSE160370 |
| **Experimental models: Organisms/strains** | | |
| *Drosophila melanogaster:* P{w[+mC]= UAS-p35.H}BH3, w[*] | Bloomington DSC | RRID:BDSC_6298 |
| *Drosophila melanogaster:* w[1118]; Dp(1;3)DC335, PBac{y[+mDint2] w[+mC]=DC335}VK00033 | Bloomington DSC | RRID:BDSC_30439 |

*(Continued on next page)*

*Continued*

| REAGENT or RESOURCE | SOURCE | IDENTIFIER |
|---|---|---|
| *Drosophila melanogaster:* w[*]; P{y[+t7.7] w[+mC]=10XUAS-IVS-mCD8::GFP}attP2 | Bloomington DSC | RRID: BDSC_32185 |
| *Drosophila melanogaster:* w[*]; P{y[+t7.7] w[+mC]=10XUAS-IVS-myr::GFP}attP40 | Bloomington DSC | RRID: BDSC_32198 |
| *Drosophila melanogaster:* w[1118]; P{y[+t7.7] w[+mC]=10XUAS(FRT.stop)GFP.Myr}su(Hw)attP1 | Bloomington DSC | RRID:BDSC_55811 |
| *Drosophila melanogaster:* w[1118]; P{y[+t7.7] w[+mC]=10xUAS(FRT.stop)myr::smGdP-V5-THS-10xUAS(FRT.stop)myr::smGdP-FLAG}su(Hw)attP5 | Bloomington DSC | RRID:BDSC_62124 |
| *Drosophila melanogaster:* w[*]; P{w[+mC]=UAS-Stinger}2, PBac{y[+mDint2] w[+mC]=13XLexAop2-IVS-tdTomato.nls}VK00022 | Bloomington DSC | RRID:BDSC_66680 |
| *Drosophila melanogaster:* w[*]; wg[Sp-1]/CyO;TI{w[+mW.hs]=lexA::VP16}fru[P1.LexA]/TM6B, Tb[1] | Bloomington DSC | RRID:BDSC_66698 |
| *Drosophila melanogaster:* w[*]; TI{FLP}fru[FLP]/TM3, Sb[1] | Bloomington DSC | RRID:BDSC_66870 |
| *Drosophila melanogaster:* TI{TI}fru[F]/TM3, Sb[1] | Bloomington DSC | RRID:BDSC_66873 |
| *Drosophila melanogaster:* TI{TI}fru[M]/T(1;3)OR60/TM3, Sb[1] Ser[1] | Bloomington DSC | RRID:BDSC_66874 |
| *Drosophila melanogaster: w[1118]; P{y[+t7.7] w[+mC]=R57C10-p65.AD}attP40; MKRS/TM6B, Tb[1]* | *Bloomington DSC* | *RRID:BDSC_70746* |
| *Drosophila melanogaster:* wg[Sp-1]/CyO; Mi{Trojan-GAL4DBD.2}Gad1[MI09277-TG4DBD.2]/TM6B, Tb[1] | Bloomington DSC | RRID:BDSC_82987 |
| *Drosophila melanogaster:* y[1] w[*]; Mi{Trojan-p65AD.0}Dbx[MI05316-Tp65AD.0]/TM6B, Tb[1] | Bloomington DSC | RRID:BDSC_82988 |
| *Drosophila melanogaster:* w[*]; TI{GAL4(DBD)::Zip-}dsx[GAL4-DBD] /TM6B, Tb[1] | Pavlou et al.[112] | N/A |
| *Drosophila melanogaster*: P{w[+mC]=13XlexAop-IVS-GFP-p10} su(Hw)attP5 | Pfeiffer et al.[113] | N/A |
| *Drosophila melanogaster:* w[*];;TI{FLP}dsx[FLP]/TM6B, Tb[1] | Rezával et al.[114] | N/A |
| *Drosophila melanogaster:* TI{GAL4}dsx[GAL4]/TM6B, Tb[1] | Rideout et al.[10] | N/A |
| *Drosophila melanogaster:* P{GAL4}odd[MC] / CyO | Gift from F. Casares | N/A |
| *Drosophila melanogaster:* TI{2A-VP16(AD)::Zip+}tsh[2A-VP16AD] | Gift from C. Desplan | N/A |
| *Drosophila melanogaster:* w[*]; UAS-myr-GFP-V5-P2A-H2B-mCherry-HA/TM3, Ser (aka UAS-WM) | Gift from J. Dubnau | N/A |
| *Drosophila melanogaster:* TI{2A-VP16(AD)::Zip+}D[2A-VP16AD] | Gift from T. Erclik | N/A |

*(Continued on next page)*

*Continued*

| REAGENT or RESOURCE | SOURCE | IDENTIFIER |
|---|---|---|
| *Drosophila melanogaster* : Canton-S (BrandX) | Gift from J. Hall | N/A |
| *Drosophila melanogaster:* vg-p65.AD[attP2FRT2] | Gift from H. Lacin | N/A |
| *Drosophila melanogaster:* TI{2A-p65(AD)::Zip+}unc-4[2A-p65AD] | Gift from H. Lacin | N/A |
| *Drosophila melanogaster*: 10x-UAS-FRT >-IVS-mCD8::GFP-STOP-FR >-IVS-CsChrimson::tdTomato (attp40) | Gift from G. Miesenböck | Vrontou et al., 2021 |
| *Drosophila melanogaster:* w[1118];; PBac{IT.GAL4}0273-G4 | Gift from S. Waddell | Gohl et al., 2011 |
| *Drosophila melanogaster:* y[1] w[*]; TI{Trojan-p65AD.1}bsh[CR00771-Tp65AD.1]/SM6a | This study | N/A |
| *Drosophila melanogaster:* y[1] w[*];; TI{Trojan-p65AD.0}CG15537[CR02107-Tp65AD.0]/TM3, Sb[1] Ser[1] | This study | N/A |
| *Drosophila melanogaster:* y[1] w[*];; Mi{Trojan-p65AD.1}CG42458[MI06870-Tp65AD.1]/TM3, Sb[1] Ser[1] | This study | N/A |
| *Drosophila melanogaster:* y[1] w[*];; Mi{Trojan-p65AD.2}Fer2[MI09483-Tp65AD.2]/TM3, Sb[1] Ser[1] | This study | N/A |
| *Drosophila melanogaster:* y[1] w[*];; Mi{Trojan-p65AD.2}fru[MI01523-Tp65AD.2]/TM3, Sb[1] Ser[1] | This study | N/A |
| *Drosophila melanogaster:* y[1] w[*];; Mi{Trojan-GAL4DBD.2}fru[MI01523-TG4DBD.2]/TM3, Sb[1] Ser[1] | This study | N/A |
| *Drosophila melanogaster:* y[1] w[*]; Mi{Trojan-p65AD.1}inv[MI09433-Tp65AD.1]/SM6a | This study | N/A |
| *Drosophila melanogaster:* y[1] w[*];;; TI{Trojan-p65AD.2}sv[CR00370-Tp65AD.2]/ci[D] | This study | N/A |
| *Drosophila melanogaster:* y[1] w[*]; Mi{Trojan-p65AD.0}Vmat[MI07680-Tp65AD.0]/SM6a | This study | N/A |
| *Drosophila melanogaster:* y[1] w[*]; Mi{Trojan-p65AD.1}Dh31-R[MI10974-Tp65AD.1]/SM6a | This study | N/A |
| *Drosophila melanogaster:* y[1] w[*];; Mi{Trojan-p65AD.0}TfAP-2[MI04611-Tp65AD.0]/TM3, Sb[1] Ser[1] | This study | N/A |
| *Drosophila melanogaster:* y[1] w[*];; Mi{Trojan-p65AD.1}Ptx1[MI11305-Tp65AD.1]/TM3, Sb[1] Ser[1] | This study | N/A |
| *Drosophila melanogaster:* y[1] w[*]; Mi{Trojan-GAL4DBD.1}dve[CR70543-TG4DBD.1]/SM6a | This study | N/A |
| *Drosophila melanogaster:* y[1] w[*]; Mi{Trojan-p65AD.1}twit[MI06552-Tp65AD.1]/SM6a | This study | N/A |

*Continued*

| REAGENT or RESOURCE | SOURCE | IDENTIFIER |
|---|---|---|
| **Software and algorithms** | | |
| R Statistical Software | R Core Team | RRID:SCR_001905 |
| RStudio | RStudio | RRID:SCR_000432 |
| Seurat (v4.1.0) | Hao et al.[115] | RRID:SCR_016341 |
| dplyr (v1.0.5) | Wickham et al.[116] | RRID:SCR_016708 |
| tidyr (v1.1.3) | Wickham[117] | RRID:SCR_017102 |
| stringr (v1.4.0) | Wickham[118] | RRID:SCR_022813 |
| ggplot2 (v3.3.3) | Wickham[119] | RRID:SCR_014601 |
| tibble (v3.1.1) | Müller and Wickham,[120] | RRID:SCR_026493 |
| cowplot (v1.1.1) | Wilke[121] | RRID:SCR_018081 |
| ggpubr (v0.4.0) | Kassambara[122] | RRID:SCR_021139 |
| pheatmap (v1.0.12) | Kolde[123] | RRID:SCR_016418 |
| ComplexHeatmap (v2.6.2) | Gu et al.[124] | RRID:SCR_017270 |
| Patchwork (v1.1.1) | Pedersen[125] | RRID:SCR_000072 |
| harmony (v1.0) | Korsunsky et al.[126] | RRID:SCR_022206 |
| AnnotationDbi (v1.52.0) | Pagès et al.[127] | RRID:SCR_023487 |
| clusterProfiler (v3.18.1) | Yu et al.[128] | RRID:SCR_016884 |
| DESeq2 (v1.30.1) | Love et al.[129] | RRID:SCR_015687 |
| Shiny | Chang et al.[130] | RRID:SCR_001626 |
| ShinyCell | Ouyang et al.[131] | RRID:SCR_022756 |
| AUCell | Aibar et al.[132] | RRID:SCR_021327 |
| pySCENIC (v0.11.2) | Van de Sande et al.[133] | RRID:SCR_025802 |
| Ruffus (v2.8.4) | Goodstadt[134] | RRID:SCR_022196 |
| cgat-core (v0.6.7) | Sims et al.[135] | RRID:SCR_006390 |
| Cytoscape | Shannon et al.[136] | RRID:SCR_003032 |
| Fiji | https://fiji.sc/ | RRID:SCR_002285 |
| Adobe Illustrator CC | Adobe Systems, San Jose, CA | RRID:SCR_010279 |
| CMTK Registration Toolkit | https://github.com/jefferis/fiji-cmtk-gui | RRID:SCR_002234 |
| FlyBase | Jenkins et al.[137]; http://flybase.org/ | RRID:SCR_006549 |
| FlyLight | HHMI Janelia | https://www.janelia.org/project-team/flylight |
| FlyWire | Dorkenwald et al.[56]; Schlegel et al.[55] | https://flywire.ai/ |
| Python | Python Software Foundation | https://www.python.org |
| VVDviewer | Wan et al.[138] Lillvis et al.[139] | https://github.com/JaneliaSciComp/VVDViewer |
| corrr (v0.4.3) | Kuhn et al.[140] | N/A |
| future (v1.21.0) | Bengtsson[141] | N/A |
| ggcorrplot (v0.1.3) | Kassambara[142] | N/A |
| Monocle3 (v1.0.0) | Cao et al.[143] | N/A |
| org.Dm.eg.db (v3.12.0) | Carlson[144] | N/A |
| readr (v1.4.0) | Wickam and Hester[145] | N/A |
| SeuratObject (v4.0.4) | Satija et al.[146] | N/A |
| SingleCellExperiment (v1.12.0) | Amezquita et al.[147] | N/A |
| SummarizedExperiment (v1.20.0) | Morgan et al.[148] | N/A |
| zinbwave (v1.12.0) | Risso et al.[149] | N/A |
| zoo (v1.8-9) | Zeileis and Grothendieck[150] | N/A |
| **Other** | | |
| Leica SP5 | Leica | RRID:SCR_018714 |

## EXPERIMENTAL MODEL AND STUDY PARTICIPANT DETAILS

### Drosophila stocks

All *Drosophila melanogaster* stocks were reared at 25°C and 40-50% humidity on standard cornmeal-agar food with a 12:12 light/dark cycle. Genotypes of the flies used were reported in the figure and legend. All strains used in the study are indicated in the key resources table.

### Full genotype list

| Figure | Full genotype |
|---|---|
| 2H, S3C | $yw^*/w^*$; $10xUAS\text{-}IVS\text{-}myr::GFP/inv^{p65AD}$; $dsx^{DBD}/+$<br>$w^*/unc\text{-}4^{AD}$;$10xUAS\text{-}IVS\text{-}myr::GFP/+$; $dsx^{DBD}/DC335$<br>$yw^*/w^*$; $10xUAS\text{-}IVS\text{-}myr::GFP/bsh^{p65AD}$; $dsx^{DBD}/+$<br>$yw^*/w^*$; $10xUAS\text{-}IVS\text{-}myr::GFP/+$; $dsx^{DBD}/Fer2^{p65AD}$<br>$yw^*/w^*$; $10xUAS\text{-}IVS\text{-}myr::GFP/+$; $dsx^{DBD}/+$; $sv^{p65AD}/+$<br>$yw^*/w^*$; $10xUAS\text{-}IVS\text{-}myr::GFP/+$; $dsx^{DBD}/TfAP\text{-}2^{p65AD}$ |
| 2N | $UAS\text{-}mCD8::GFP/+$; $dsx^{DBD}$, $UAS\text{-}mCD8::GFP/+$ |
| 3F, S7E | $yw^*/w^*$; $VGlut^{DBD}/+$; $10xUAS\text{-}IVS\text{-}mCD8::GFP/Ptx1^{p65AD}$ |
| 3K, S7F | $yw^*/w^*$; $dve^{DBD}/UAS\text{>}mCD8::GFP\text{>}Chrimson::tdTomato$; $fru^{FLP}/Ptx1^{p65AD}$ |
| 4E,G, S8A | $yw^*/w^*$; $10xUAS\text{-}IVS\text{-}mCD8::GFP/+$; $Gad1^{DBD}/TfAP\text{-}2^{p65AD}/+$ |
| 4F, S8A,B | $w^*/w^*$; $UAS\text{>}stop\text{>}myrGFP/+$; $Gad1^{DBD}$, $fru^{FLP}/TfAP\text{-}2^{p65AD}$ |
| 4I, S8C | $w^*/w^*$; $13xlexAop\text{-}IVS\text{-}GFP\text{-}p10/+$; $fru^{P1.LexA}/fru^{F}$<br>$w^*/w$; $13xlexAop\text{-}IVS\text{-}GFP\text{-}p10/+$; $fru^{P1.LexA}/fru^{M}$ |
| 4J, S8E | $UAS\text{-}p35$, $w^*/w^*$; $10xUAS\text{-}IVS\text{-}mCD8::GFP/+$; $Gad1^{DBD}/TfAP\text{-}2^{p65AD}$ |
| 6D, S12A | $yw^*/w^*$; $UAS\text{>}stop\text{>}myr::GFP/+$; $dsx^{DBD}$, $Fer2^{p65AD}/fru^{FLP}$<br>$yw^*/w^*$; $10xUAS\text{-}IVS\text{-}myr::GFP/+$; $dsx^{DBD}/CG42458^{p65AD}$<br>$yw^*/w^*$; $10xUAS\text{-}IVS\text{-}myr::GFP/+$; $dsx^{DBD}/Dbx^{p65AD}$<br>$yw^*/w^*$; $10xUAS\text{-}IVS\text{-}myr::GFP/+$; $dsx^{DBD}/CG15537^{p65AD}$ |
| S4H | $Y/w^*$; $UAS\text{-}WM/Vmat^{p65AD}$; $fru^{DBD}/+$ |
| S6A | $Y/w^*$; $UAS\text{-}WM/+$; $fru^{DBD}/+$; $sv^{p65AD}/+$ |
| S6B | $Y/w^*$; $10xUAS\text{-}IVS\text{-}myr::GFP/+$; $dsx^{DBD}/+$; $sv^{p65AD}/+$ |
| S6D | $Y/unc\text{-}4^{p65AD}$; $UAS\text{-}WM/+$; $fru^{DBD}/DC335$ |
| S6F | $Y/w^*$; $UAS\text{-}WM/+$; $fru^{DBD}/TfAP\text{-}2^{p65AD}$ |
| S7D | $w^*/w^*$; $VGlut^{DBD}/UAS\text{>}stop\text{>}myr::GFP$; $fru^{FLP}/Ptx1^{p65AD}$ |
| S7G | $Y/w^*$; $UAS\text{-}WM/+$; $fru^{DBD}/Fer2^{p65AD}$<br>$Y/w^*$; $odd^{Gal4}/10xUAS\text{>}stop\text{>}myr::smGdP\text{-}V5\text{-}THS\text{-}10xUAS\text{>}stop\text{>}myr::smGdP\text{-}FLAG$; $fru^{FLP}/+$<br>$Y/w^*$; $UAS\text{-}WM/+$; $fru^{DBD}/TfAP\text{-}2^{p65AD}$<br>$Y/w^*$; $UAS\text{-}WM/+$; $fru^{DBD}/Ptx1^{p65AD}$<br>$Y/unc\text{-}4^{p65AD}$; $UAS\text{-}WM/+$; $fru^{DBD}/DC335$<br>$Y/w^*$; $UAS\text{-}WM/+$; $fru^{DBD}/Dbx^{p65AD}$<br>$Y/w^*$; $UAS\text{-}WM/bsh^{p65AD}$; $fru^{DBD}/+$<br>$Y/w^*$; $UAS\text{-}WM/inv^{p65AD}$; $fru^{DBD}/+$<br>$Y/w^*$; $UAS\text{-}WM/+$; $fru^{DBD}/D^{VP16AD}$<br>$Y/w^*$; $UAS\text{-}WM/+$; $fru^{DBD}/+$; $sv^{p65AD}/+$<br>$Y/w^*$; $UAS\text{-}WM/vg^{p65AD}$; $fru^{DBD}/+$<br>$Y/w^*$; $UAS\text{-}WM/tsh^{VP16AD}$; $fru^{DBD}/+$<br>$Y/w^*$; $UAS\text{-}WM/twit^{p65AD}$; $fru^{DBD}/+$<br>$Y/w^*$; $UAS\text{-}WM/dve^{p65AD}$; $fru^{DBD}/+$ |
| S8D | $w^*/w^*$; $13xlexAop\text{-}IVS\text{-}GFP\text{-}p10/+$; $fru^{P1.LexA}/+$ |
| S8F | $yw^*/w^*$; $10xUAS\text{-}IVS\text{-}myr::GFP/Dh31\text{-}R^{p65AD}$; $dsx^{DBD}/+$<br>$Y/w^*$; $UAS\text{-}WM/Dh31\text{-}R^{p65AD}$; $fru^{DBD}/+$ |
| S10A,B | $Y/w^*$; $UAS\text{-}WM/+$; $fru^{DBD}/+$; $sv^{p65AD}/+$ $yw^*/w^*$;<br>$10xUAS\text{-}IVS\text{-}myr::GFP/+$; $dsx^{DBD}/+$; $sv^{p65AD}/+$ |

Sexes of all genotypes are stated in the figures.

**CellPress**

## METHOD DETAILS

### Generation of meta-central brain neuronal atlas

The generation of this atlas is described in detail in a companion paper.[19] All code and specific details of what was run, please refer to https://github.com/aaron-allen/Dmel-adult-central-brain-atlas. Briefly, we combined our newly generated scRNA-seq data of the adult central brain[19] with multiple re-processed publicly available 10x chromium 3' chemistries datasets.[102–109,111,151] These datasets were integrated, and non-neuronal cells, peripheral neurons, and optic lobe neurons were annotated and removed. The remaining central brain neurons were re-clustered, resulting in a well-integrated transcriptional central brain neuron meta-atlas with 329,466 cells.

### Annotation of cell types and estimate of depth of coverage

Broad neurotransmitter identities (cholinergic, glutamatergic, GABAergic, monoaminergic, and neuroendocrine), as well as select specialized cell types (Kenyon cells and motor neurons), were annotated using curated sets of marker genes. For each class, we computed module scores across all cells using Seurat's "AddModuleScore" function. To ensure comparability across gene programs, these scores were l2-normalized, a method that scales each score relative to the magnitude of the cell's expression profile by dividing by the square root of the sum of squared values. This normalization ensures comparability across cells and modules by accounting for differences in gene expression magnitude and sequencing depth. A cell was annotated to a given class if it had a positive module score for that gene program and that score exceeded all others. Cluster-level annotations were assigned based on the majority identity within each cluster (resolution 10). To refine granularity, each annotated class was separately extracted, re-integrated, and re-clustered to identify transcriptionally distinct subtypes. These refined annotations were then mapped back onto the full meta-central brain atlas.

Cluster robustness was assessed through comparison with anatomical marker genes and available split-GAL4 driver lines, as described in detail in the companion paper.[19] Throughout the manuscript, we refer to clusters as "cell types" for consistency, while acknowledging that some transcriptional types may still contain functional or connectivity heterogeneity. Overall, this annotation pipeline supports a hierarchical and biologically grounded classification scheme, providing a robust framework for exploring sex-specific differences at the level of transcriptionally defined neuronal types.

To estimate the cellular depth of coverage in our atlas, we compared transcriptomic cell counts for multiple anatomically defined neuronal populations to corresponding estimates from the Flywire central brain connectome.[55,56] These comparisons spanned populations ranging from small clusters of ~4 neurons to broad classes encompassing thousands of cells. Depth of coverage was calculated as the ratio of cells in our dataset to the expected anatomical number for each population. Across all populations examined, this yielded an average depth of 9.8× and exhibited perfect correlation with anatomical benchmarks (R = 1; see[19]).

### Annotation of sex

Two of the datasets used in our meta-analysis did not separate the sexes into separate samples.[103,105] As a result, we've had to rely on in-silico sexing of these samples. We used Seurat's "AddModuleScore" function with the male-specific genes *lncRNA:roX1* and *lncRNA:roX2*. Cells with a module score less than zero were labeled female, and cells with module scores greater than zero were labeled male. To test the efficacy of this method, we performed *in silico* sexing of the datasets where the sexes were processed separately and achieved a precision of 0.93-1.00 and a recall of 0.89-0.97 for identifying a cell as female, depending on the dataset.

### Differential gene expression analysis

Differential gene expression between the sexes was conducted in a similar fashion to.[105] We use the zinbwave package (Zero-Inflated Negative Binomial-based WAnted Variation Extraction) to account for dropout and the count-based nature of the data.[149] We included "dataset" as a covariate in the model to correct for differences between the individual datasets. Differential expression was conducted with DESeq2.[129] Even with correcting for dataset in the models for the differential expression testing, multiple genes initially recovered to be sex-biased exhibited strong dataset-specific expression (Figure S2). To eliminate these genes, we filtered the genes by the following criteria: (1) the total UMI of the gene across all cells greater than or equal to 400, (2) the maximum UMI observed within a cell greater than or equal to 4, (3) the percent contribution of UMI from each dataset less than or equal to 60%, and (4) the percent contribution of UMI from the Park et al. (2025)[109] dataset less than or equal to 30%. Genes fitting these criteria were removed from all differential gene expression analyses shown in this manuscript.

### Gene ontology analysis

Gene ontology analysis was performed using the packages "clusterProfiler"[152] with "org.Dm.eg.db"[153] and " AnnotationDbi".[154] The "enrichGO" function was used to determine the enriched terms for each of the molecular function, biological process, and cellular compartment categories (Figure S1E).

### Subclustering *doublesex* and *fruitless* neurons

To subcluster *dsx*+ neurons, all neurons with greater than or equal to 1 UMI of *dsx* expression were extracted. For our generated dataset,[19] we transgenically expressed *EGFP* using *dsx^Gal4*, and so we also included any cell with at least 1 UMI of *EGFP* from

our dataset. For *fru+* subcluster, dataset specific thresholds were chosen (Davie-2018 > 1; Baker-2021 > 2; Park-2022 > 2; Dopp-2024 > 3; LeeBenton-2023 > 3; Li-2022 > 3; Lu-2023 > 3; Park-2025 > 3). These cells were extracted and re-integrated using the "harmony" package[155] using "RunHarmony", correcting for dataset of origin, preparation type (cell vs. nuclei), and individual sample ID metadata. The re-integrated cells were clustered with "FindNeighbors" and "FindClusters" using the Louvain algorithm. We note that although our dataset achieves 9.8× estimated coverage, it may still underrepresent rare subtypes, and additional sampling may be needed to resolve the full diversity observed in morphological reconstructions of *dsx+* and *fru+* neurons.[55,56] Indeed, we currently have not been able to reliably identify all sex-specific *dsx+* cell types with only one cell per hemisphere (data not shown).

### Early- vs. late-born annotation

To annotate early-born ("Imp>dati") and late-born ("dati>Imp"), we used Seurat's "AddModuleScore" to calculate expression scores for two established temporal transcriptional programs: the early markers *Imp* and *mamo*, and the late markers *dati* and *pros*.[156–160] These module scores were then l2-normalized, scaling the vector for each gene program to a magnitude of 1. Using these normalized scores, we annotated neurons as early-born if their early-born score was both greater than zero and greater than their late-born score. Conversely, neurons were considered late-born if their late score met both criteria.

### Gene regulatory network analysis with SCENIC

Gene regulatory network analysis was performed as previously described.[19] Briefly, pySCENIC[133] (v0.11.2) was run on select, individual subclustered clusters to identify transcription factor-based regulons by using the gene expression data as input. The raw expression matrix was filtered, modules comprising transcription factors and co-expressed genes were generated using GRNBoost2, and then pruned to remove indirect targets lacking transcription factor motif using cisTarget. Cells were then scored for the activity of each high-confidence regulon using the AUCell.[132] Cell-type enriched regulons were calculated using Seurat's "FindAllMarkers" function. Gene regulatory networks were visualized with Cytoscape (Shannon et al., 2003). For visualization purposes, GRNs were further filtered to only include positive regulons less than 100 genes in size, and each node (gene) had to have a maximum expression of at least 3 UMI and total expression of at least 80 UMI.

### Pseudotime analysis

Pseudotime analysis was performed using Monocle3[143] (v1.0.0) to infer developmental trajectories from our single-cell RNA-seq data using standard workflows. In brief, a *CellDataSet* was created using "new_cell_data_set", followed by preprocessing and alignment ("preprocess_cds", "align_cds"). Dimensionality reduction and clustering were performed using "reduce_dimension" and "cluster_cells, and trajectories were learned using "learn_graph" using default settings. To define the developmental starting point, we used "order_cells" with *Imp+* cells (early-born marker) set as the root of the trajectory. This allowed us to order all cells along a continuous pseudotime axis representing their inferred developmental progression. We identified genes whose expression varied significantly across pseudotime using "graph_test" and filtered for q-value less than 1e-50 and Moran's I greater than 0.2, indicating strong spatial autocorrelation along the trajectory. Heatmaps of significantly varying genes were plotted with the package "ComplexHeatmap".[161]

### Re-clustering and correcting for early- vs. late-born neurons

In our companion manuscript,[19] we show that early-born *Imp+* neurons have separated "punctate" like clustering as compared to the late-born *dati+* neurons. We also showed that these late-born unsupervised clusters may correspond to hemilineages. To better represent full hemilineages and include the appropriate early-born and late-born neurons with unsupervised clusters, we projected early- vs late-born into a shared embedding using the "harmony" package. This re-clustered embedding was used to identify and extract hemilineage representations (Figures 3I; 3J; 4K and 5F).

### Generating a web app

Our interactive web tool for visualization of these scRNA-seq data was generated using a modified version of "ShinyCell", an R-based Shiny App[131]; https://github.com/SGDDNB/ShinyCell). Specifically, we forked and modified "easyshiny" (https://github.com/NBISweden/easyshiny), which is a forked version of the original "ShinyCell". Our modified version is available here - https://github.com/aaron-allen/easyShinyCell.

### Generation of split-Gal4 driver lines

We used existing coding intronic Minos-mediated integration cassette/CRISPR-mediated integration cassette (MiMIC/CRIMIC) lines[162,163] (see Key Resources Table) to generate split-GAL4 drivers using the Trojan method.[164] pBS-KS-attB2-SA()-T2A-Gal4DBD-Hsp70 or pBS-KS-attB2-SA()-T2A-p65AD-Hsp70 vectors[164] with the appropriate reading frame were inserted in the MIMIC/CRIMIC locus of a given line via recombinase-mediated cassette exchange through injection (BestGene, CA). Stocks were generated with the transformed flies as described in.[164]

## Immunohistochemistry

After a brief pre-wash of adult flies in 100% EtOH to remove hydrophobic cuticular chemical compounds, brains were dissected in PBS at RT (20-25°C), collected in 2 mL sample tubes and fixed with 4% formaldehyde (Sigma-Aldrich) in PBS (Sigma-Aldrich) for 20 min at RT. After fixation, tissues were washed in 0.5-0.7%PBS/Triton X-100 (Sigma-Aldrich) (PBT) 3 times each for 20 min at RT. After blocking in 10% normal goat serum (Sigma-Aldrich) in PBT (NGS/PBT) overnight (8-12 h) at RT, tissues were incubated in primary antibody solutions for 48-72 hrs at 4°C (1:1000, rabbit anti-GFP, Thermo Fisher Scientific; 1:1000, chicken anti-GFP, Abcam; 1:1000, rabbit anti-RFP, Antibodies Online; 1:10, mouse anti-Brp, Developmental Studies Hybridoma Bank; 1:500, rabbit anti-Dh31, Lin et al., 2022; 1:500, guinea pig anti-Otp, Hildebrandt et al., 2020[101]). After 4 washes in PBT for 1 h each at RT, tissues were incubated in secondary antibody solutions for 48 h at 4°C with the exception of anti-Dh31 staining, in which tissues were incubated in secondary antibody solution for 24 h at 4°C (1:500, anti-rabbit Alexa Fluor 488, anti-rabbit Alexa Fluor 633, anti-chicken Alexa Fluor 488, anti-mouse Alexa Fluor 546, anti-rat Alexa Fluor 633, anti-guinea pig Alexa Fluor 633, Thermo Fisher Scientific). After 4 washes in PBT for 1 h each at RT specimens were imaged directly or 70% glycerol in PBS was added to the sample tubes, which were subsequently transferred to -20°C and kept for at least 8 hr for tissue clearing. Specimens were mounted in Vectashield (Vector Laboratories).

## Confocal image acquisition and processing

Confocal image stacks were acquired on a Leica TCS SP5 confocal microscope at 1024 1024-pixel resolution with a slice size of 0.29 μm or 1 μm. Water-immersion 25x and oil-immersion 40x objective lenses were used for brain images. Images were registered onto the unisex template brain using the Fiji Computational Morphometry Toolkit (CMTK) Registration GUI (https://github.com/jefferis/fiji-cmtk-gui). For segmented images, we used the software VVDViewer[138,139] (https://github.com/JaneliaSciComp/VVDViewer) to render the registered image stacks in 3D, manually mask other neurons co-labeled in the image, and segment out neurons of interest. All segmented images include unsegmented versions within the supplemental figures.

