## [Document S2. Transparent peer review records for Allen et al. · Cell Genomics]

**Differential Neuronal Survival Defines a Novel Axis of Sexual Dimorphism in the
Drosophila brain**

Aaron M. Allen, Megan C. Neville, Tetsuya Nojima, Faredin Alejevski, and Stephen F. Goodwin

Summary

Initial submission: Received : Jun 27, 2025

Scientific editor: Sara Rohban

First round of review: Number of reviewers: 2
Revision invited : Jul 31, 2025
Revision received : Oct 31, 2025

Second round of review: Number of reviewers: 1
Accepted : Dec 08, 2025

Data freely available: YES

Code freely available: YES

This transparent peer review record is not systematically proofread, type-set, or edited. Special characters, formatting, and equations may fail to render properly. Standard procedural text within the editor's letters has been deleted for the sake of brevity, but all official correspondence specific to the manuscript has been preserved.

Referees' reports, first round of review

Reviewer #1:

This manuscript describes a thorough investigation of sexual dimorphism in the *Drosophila* brain at single-cell resolution. The analyses are based on data from multiple single-cell studies, resulting in a very large dataset where even rare subtypes of central brain neurons are well represented. This allows the authors to explore cell type diversity at high resolution and to characterize sexually dimorphic gene expression in narrowly defined cell types. Transcriptomic analyses are carefully validated by the imaging of specific cell populations / neuronal subtypes using split-Gal4 markers. The results confirm much that was already known: in particular, that pronounced sexual dimorphism is limited to neurons that express the transcription factors *dsx* and/or *fru*, and that some neurons express both *dsx* and *fru* while others express only one or the other. The finding that different TFs, neurotransmitters, transmembrane receptors, and other genes show sex-biased expression in different cell types is expected. However, knowing the exact identities of these genes sets the stage for characterizing and experimentally manipulating the functions of well-defined populations of sexually dimorphic neurons. The most novel results concern the developmental origin of sex-specific neurons. The authors present evidence that key populations of male- and female-specific neurons come from different cell sub-lineages that differ in the timing of their birth in the developing brain. The authors suggest that sexual dimorphism in the brain emerges from sex-specific proliferation and pruning of cell lineages, rather than from sex-specific differentiation of the same cell lineages in males vs females. This conclusion changes our understanding of the neural substrate of sex-specific behavior, and should stimulate work in other model systems. However, this side of the story is the least clearly described, which is my main concern about this manuscript - see below.

What I found especially appealing about this paper is how it starts with a completely unbiased transcriptomic approach, and produces results that mesh very well with neuroanatomy and genetics. Early on, the authors identify sexually dimorphic neuronal populations without marking them either genetically or bioinformatically (by subsetting) - then they show, after the fact, that these populations express the genes that control sexual dimorphism in the brain. Later, they show that many subpopulations of cells defined by unsupervised clustering of their transcriptomes correspond to previously known groups of neurons defined anatomically and/or genetically. In addition to serving as the best possible validation of the transcriptomic results, this unbiased strategy also led to the identification of new genes with previously unappreciated roles in sex-specific brain development (e.g., *otp*), as well as to the discovery of previously unknown classes of sex-specific neurons. This aspect of the paper is really beautiful.

But in another respect, I found this paper frustrating to read. Until one gets to the Discussion (which I found very clear and engaging), the paper often reads like a technical report or resource description, with the big take-home messages lost among the details. The manuscript assumes that the reader (1) is very familiar with the details of *Drosophila* brain development and (2) finished reading the companion paper 5 minutes ago. (I did not see the companion paper,

which likely colors my impression of this one). To convey the key messages of this paper to a less than expert reader, I feel that some background and better explanation are needed.

Specifically, the most novel conclusions from this work are framed in the context of cell lineage and developmental timing. It is from the superposition of lineage, timing, and gene expression that the interesting lessons emerge. However, this paper does not even explain what hemilineages are. I'm sure it's explained somewhere in that companion paper - but without this background, it's very hard to make sense of this one. I recommend adding some background explaining: how hemilineages develop; which lineages are early- vs late-born, and how they are distinguished; which TFs define the key early and late lineages; and how *dsx* and *fru* expression maps onto these lineages. This background information should be explained either at the beginning of the Results text, or at least before the authors begin to discuss their finding in terms of lineage and timing. In addition, I suggest adding some schematics introducing early and late cell lineages to Figure 5. That would really help the readers interpret the data in Figures 5-7 and the associated Results text.

On a related note - in the Methods section of the Supplement (lines 1069-1076), I think it is necessary to describe in more detail how early- and late-born cells were identified, and why that particular approach was used. Similarly, pseudotime analysis (lines 1092-1100) also needs to be explained better. Both of these approaches are key to understanding the authors' results, including those featured more prominently in the Abstract and Discussion. And please explain somewhere what is meant by "hemilineage approximation", "extracted approximations of hemilineages", etc. - it would be good to know how this "approximation" is done, and how close it is.

Minor comments.

I really don't like the word "exaptation" in the title. For one thing, inferring exaptation would require comparative data, which this paper lacks, and the concept of exaptation is only tenuously related to the ideas that the authors present in their Discussion. And on the flip side, this title does not reflect any of the specific, and often quite interesting inferences that the Discussion contains. Please consider a different title.

Line 104 - the calculation of cellular depth of coverage is not explained anywhere in the paper. Please explain it in the Methods section, with a very brief summary in the Results. Otherwise, it's not clear what a particular combination of the total number of cells and the cellular depth of coverage implies, or how one should interpret differences in the size of neuronal populations (for example, *dsx+* vs *fru+* neurons). Cellular depth is referred to at multiple points in the text and figures (for example, "2108 cells with 9.8-fold depth of coverage" when discussing *dsx+* neurons). In addition to explaining how this depth is calculated, please be clearer throughout the paper when the text refers to the cells based on imaging, or the cells in UMAP clusters (e.g., line 288-289).

Line 103 - the phrasing, "only a few [cell types] exhibiting dramatic sex differences in the number of genes differentially expressed" is hard to interpret; "differences" between what and what? Please rephrase.

Fig 1C - since the point is to illustrate sexual dimorphism, it might be better to intersperse male and female lanes instead of grouping them separately by sex.

Fig 1E - given the discussion in the text, it would be helpful to see a similar volcano plot after excluding dsx+ and fru+ cells.

Fig 1F legend - the meaning of "shared between the sexes (albeit in different cell types)" is hard to figure out, please rephrase.

Fig 1E legend - was testing performed separately in each cell type?

Fig 1H - it could be helpful to label a few selected "interesting" cell types on the left (I assume they were all identified in retrospect?). I also noticed that a couple of cell types express dsx and fru but do not show differences in abundance or gene expression - what are those?

Fig 2B, C - please explain how coverage was estimated.

Fig 2K legend - is the left graph males only?

Fig 3D - does the dashed box correspond to Fig 3E?

Fig 3H - why are some genes in bold? Transcription factors?

Fig 3I and Fig 4L - please mark early/late on the pseudotime bar (after that it should clear).

Figure 5 - this would be an excellent place to introduce the background schematics of hemilineages and their timing.

Fig 6D - is there anything that defines subtype 7?

Fig 6H legend - please mention what the arrowhead designates.

Fig 7D-F - it's very hard to make sense of these data. All that can be seen is that some cell types express both early and late-born genes, while others are enriched for one or the other; and that most cell types clearly enriched for late or early genes are sexually monomorphic. The key point of how earliness/lateness relates to sexual dimorphism is not obvious from these figures. Could showing a more selective subset of the data help?

Fig 7G - I found this way of representing the data incredibly confusing. I suspect a bar chart would be easier to interpret.

Line 1048 - I assume that, after these corrections, genes inferred to be sex-specific no longer exhibited strong dataset-specific expression - but this is not explicitly stated, and the post-correction data are not shown in Fig S2.

Line 1071 - please explain what "I2-normalised" score means, and why it is being used.

Reviewer #2:

The authors use single-cell transcriptomics to profile Fruitless and Doublesex expressing neurons throughout the central brain (excluding the optic lobes) - classifying sex-specific or sexually dimorphic cell types in males and females. This is an important study that complements recently released connectomes (hemibrain and FlyWire) for the female brain and a forthcoming male brain connectome (the comparisons of which can be used to define sex-specific and sexually dimorphic cell types based on connectivity, but connectome-based comparisons won't reveal differences in gene expression between cell types, what is covered here). Importantly, despite a number of studies on Fruitless and Doublesex expressing neurons in *Drosophila* over the past decade, we still know relatively little about the transcriptional differences that define Fru and Dsx cell types - this paper is an important advance towards that goal. The authors validate

their findings by building split-GAL4 lines based on the results and showing that distinct combinations of gene expression define morphological cell types - these driver lines will be useful to the field. And, for some examples (like the Dh31-expressing mAL neurons), the authors make manipulations to determine why particular genes are expressed in male Fru neurons but not female Fru neurons. They identify two genes (Imp and dati) that mark early and late born neurons respectively, and relate this gene expression to female and male specific dsx/fru neurons, discovering that what sets the female brain apart from the male brain is when in development PCD (programmed cell death) occurs. Finally, the data are presented very well (figures are very easy to follow) and the writing is clear. I have very few concerns with the manuscript and am overall supportive of publication - my comments below should be addressed prior to publication.

Major Issues:

Fig. 1 - I could not find information on how neurons sequenced are sorted into cell types. Presumably there is some heterogeneity (both transcriptionally and in terms of synaptic connectivity (e.g., see Schlegel et al. and Matsliah et al. Nature 2024 papers - another, more exhaustive, way to define cell types)) within what is called a cell type here - how much? Cell types are critical to this paper - they are the fundamental unit used for making comparisons between male and female central brains - how they are defined (and what sources of error there are in defining them) needs to be laid out a bit more. Some of this may be presented in the companion paper (which I did not review) - would be helpful to a reader to have this information here.

Fig. 2 - see Deutsch et al. bioRxiv 2025 - there are 16 subtypes of Dsx+ neurons based on morphology and connectivity in the female brain (the number in the male brain is not yet known because the male brain connectome has not been released) - this study (Allen et al.) reports 6 subpopulations but refers to them as cell types. This is a bit confusing, and some explanation (again related to the definition of a cell type) would be helpful. As just one example, pC1 (covered in more depth in Fig. 6) is a cell type in this figure. But in female connectomes, pC1a-e are distinct cell types (again, based on morphology and connectivity) that drive different behaviors (e.g., receptivity versus aggression). Presumably pC1a-e have different transcriptional profiles - can those not be detected here (it looks like there might be clusters within the pC1 type in females in Fig 6)? If the 9.8x coverage is not high enough to detect differences between the pC1 subtypes (as just one example), make that clear. A similar issue also pertains to Fig. 3.

Could not evaluate the underlying data (or comment on how it will be shared) at flycns.com - the website indicates it is 'coming soon'

Minor Issues/Other Comments:

Introduction - Line 69-70 does not make sense - the brain does not integrate sensory inputs, internal states, and behavioral outputs (or perhaps I don't understand the authors' meaning) - rephrase to indicate that the central brain of *Drosophila* (not including the optic lobes) integrates sensory information and drives behavioral outputs, both relative to internal states.

Line 103 - Can you justify the 9.8x coverage - what will be missed at this coverage level?

Line 138 - How is subtle defined? Given that the expectation is that only a fraction of the cell

types in the central brain is sex-specific or sexually dimorphic, aren't the results consistent with that expectation? Also wouldn't the biggest differences in gene expression be during pupal development, not in adults?

Discussion - would be helpful to readers to discuss the results presented here in the context of published and emerging connectomes - connectivity is a complementary (and more exhaustive) way to define cell types - see for example Dombrovski et al. Nature 2025 or Yoo et al. CB 2023
The authors report that Fru is expressed in 57% of hemilineages - what is special about the hemilineages that do not express Fru?

Authors' response to the first round of review

RESPONSE TO REVIEWERS

Reviewer #1: This manuscript describes a thorough investigation of sexual dimorphism in the *Drosophila* brain at single-cell resolution. The analyses are based on data from multiple single-cell studies, resulting in a very large dataset where even rare subtypes of central brain neurons are well represented. This allows the authors to explore cell type diversity at high resolution and to characterize sexually dimorphic gene expression in narrowly defined cell types. Transcriptomic analyses are carefully validated by the imaging of specific cell populations / neuronal subtypes using split-Gal4 markers. The results confirm much that was already known: in particular, that pronounced sexual dimorphism is limited to neurons that express the transcription factors *dsx* and/or *fru*, and that some neurons express both *dsx* and *fru* while others express only one or the other. The finding that different TFs, neurotransmitters, transmembrane receptors, and other genes show sex-biased expression in different cell types is expected. However, knowing the exact identities of these genes sets the stage for characterizing and experimentally manipulating the functions of well-defined populations of sexually dimorphic neurons. The most novel results concern the developmental origin of sex-specific neurons. The authors present evidence that key populations of male-and female-specific neurons come from different cell sub-lineages that differ in the timing of their birth in the developing brain. The authors suggest that sexual dimorphism in the brain emerges from sex-specific proliferation and pruning of cell lineages, rather than from sex-specific differentiation of the same cell lineages in males vs females. This conclusion changes our understanding of the neural substrate of sex-specific behavior, and should stimulate work in other model systems. However, this side of the story is the least clearly described, which is my main concern about this manuscript - see below.

What I found especially appealing about this paper is how it starts with a completely unbiased transcriptomic approach, and produces results that mesh very well with neuroanatomy and genetics. Early on, the authors identify sexually dimorphic neuronal populations without marking them either genetically or bioinformatically (by subsetting) - then they show, after the fact, that these populations express the genes that control sexual dimorphism in the brain. Later, they show that many subpopulations of cells defined by unsupervised clustering of their transcriptomes correspond to previously known groups of neurons defined anatomically and/or genetically. In addition to serving as the best possible validation of the transcriptomic results,

this unbiased strategy also led to the identification of new genes with previously unappreciated roles in sex-specific brain development (e.g., *otp*), as well as to the discovery of previously unknown classes of sex-specific neurons. This aspect of the paper is really beautiful.

But in another respect, I found this paper frustrating to read. Until one gets to the Discussion (which I found very clear and engaging), the paper often reads like a technical report or resource description, with the big take-home messages lost among the details. The manuscript assumes that the reader (1) is very familiar with the details of *Drosophila* brain development and (2) finished reading the companion paper 5 minutes ago. (I did not see the companion paper, which likely colors my impression of this one). To convey the key messages of this paper to a less than expert reader, I feel that some background and better explanation are needed.

We thank the reviewer for their careful and thorough evaluation of our manuscript. We are especially grateful for the recognition of the novelty and rigour of our analysis, and for highlighting the value of our unbiased transcriptomic approach. In response to the reviewer's suggestion that the developmental framework (particularly involving hemilineages and birth timing) was not clearly explained, we have made the appropriate revisions throughout the manuscript (described in detail below). We hope that these changes clarify the logic and significance of our findings, making the manuscript more accessible to readers outside the immediate field.

Specifically, the most novel conclusions from this work are framed in the context of cell lineage and developmental timing. It is from the superposition of lineage, timing, and gene expression that the interesting lessons emerge. However, this paper does not even explain what hemilineages are. I'm sure it's explained somewhere in that companion paper - but without this background, it's very hard to make sense of this one. I recommend adding some background explaining: how hemilineages develop; which lineages are early- vs late-born, and how they are distinguished; which TFs define the key early and late lineages; and how *dsx* and *fru* expression maps onto these lineages. This background information should be explained either at the beginning of the Results text, or at least before the authors begin to discuss their finding in terms of lineage and timing. In addition, I suggest adding some schematics introducing early and late cell lineages to Figure 5. That would really help the readers interpret the data in Figures 5-7 and the associated Results text.

We thank the reviewer for highlighting the need for additional developmental context. To improve clarity for readers unfamiliar with our companion study, we now include a recap of the main conclusions at the beginning of the results section. We have also added schematics to Figure 5 that illustrates the generation of hemilineages and highlights how early and late neuronal identities arise during development, providing a clearer framework for interpreting Figures 5–7 and the associated text.

On a related note - in the Methods section of the Supplement (lines 1069-1076), I think it is necessary to describe in more detail how early- and late-born cells were identified, and why that particular approach was used. Similarly, pseudotime analysis (lines 1092-1100) also needs to be explained better. Both of these approaches are key to understanding the authors' results, including those featured more prominently in the Abstract and Discussion. And please explain somewhere what is meant by "hemilineage approximation", "extracted approximations of hemilineages", etc. - it would be good to know how this "approximation" is done, and how close it is.

We appreciate the reviewer's comments and suggestions. We have now added more detail to the Methods section regarding pseudotime analysis and early- vs late-born cell identification. We have additionally added a sentence to the results which clarifies the use of birth order defining genes: "We further demonstrate that persistent expression of *Imp* and *dati* reliably marks early- and late-born neurons, respectively, providing effective proxies for temporal identity." We have also added a description of pseudotime to the results section: "across pseudotime (Figure 3I), which orders cells along a continuous trajectory based on their gene expression profiles."

Regarding hemilineage approximations, we were being (admittedly overly) cautious with our claims. To annotate these hemilineages, we used the described genetic intersections and saw that the labelled populations follow the Hemilineage Associated Tracts (HATs; Lovick et al, 2013) for each of these hemilineages. The isolated and extracted clusters that corresponded to these gene expression patterns also had consistent numbers to those seen in recent connectome annotations of these hemilineages (Schlegel et al., 2024). And so, these very likely represent the hemilineages we've annotated them as. To avoid confusion, we have now removed multiple instances of "approximation" from our descriptions in the main text and have elaborated on these details in the methods section.

Minor comments

I really don't like the word "exaptation" in the title. For one thing, inferring exaptation would require comparative data, which this paper lacks, and the concept of exaptation is only tenuously related to the ideas that the authors present in their Discussion. And on the flip side, this title does not reflect any of the specific, and often quite interesting inferences that the Discussion contains. Please consider a different title.

We appreciate your suggestion and have adjusted the title accordingly.

Line 104 - the calculation of cellular depth of coverage is not explained anywhere in the paper. Please explain it in the Methods section, with a very brief summary in the Results. Otherwise, it's not clear what a particular combination of the total number of cells and the cellular depth of coverage implies, or how one should interpret differences in the size of neuronal populations (for example, *dsx+* vs *fru+* neurons). Cellular depth is referred to at multiple points in the text and figures (for example, "2108 cells with 9.8-fold depth of coverage" when discussing *dsx+* neurons). In addition to explaining how this depth is calculated, please be clearer throughout the paper when the text refers to the cells based on imaging, or the cells in UMAP clusters (e.g., line 288-289).

We thank the reviewer for this important clarification. We have now added a description of how cellular depth of coverage was calculated to the Methods section. Specifically, we compared transcriptomic cell counts for multiple anatomically defined populations against neuron counts from the Flywire central brain connectome. The average ratio of cells recovered relative to expected counts across populations yielded a cellular depth of coverage of 9.8 \times , with perfect

correlation ($R = 1$). We have also added a summary to the Results section, when cellular depth is first mentioned, to aid reader interpretation throughout the manuscript.

Line 103 - the phrasing, "only a few [cell types] exhibiting dramatic sex differences in the number of genes differentially expressed" is hard to interpret; "differences" between what and what? Please rephrase.

We have now rephrased this sentence. "A small number of cell types, however, exhibited dramatic differences in the number of genes differentially expressed and differential abundance between the sexes."

Fig 1C - since the point is to illustrate sexual dimorphism, it might be better to intersperse male and female lanes instead of grouping them separately by sex.

We appreciate the suggestion; we had initially explored displaying the sexes interspersed. However, we found that doing so required substantially more complex labelling, which made the panel harder to interpret. Given the main message of this panel is that most of these genes are ubiquitously expressed across both sexes, we believe that grouping male and female lanes separately is clearer and more immediately legible. That said, we are happy to include an interspersed version in a supplemental figure if the review feels it would be valuable for comparison.

Fig 1E - given the discussion in the text, it would be helpful to see a similar volcano plot after excluding *dsx*⁺ and *fru*⁺ cells.

Thank you for this suggestion. We have now added a supplement figure (Figure S1) where we show the progression of all DEGs across all cell types with successive filtering of specific genes and cell types.

Fig 1F legend - the meaning of "shared between the sexes (albeit in different cell types)" is hard to figure out, please rephrase.

For clarity we now state "or found in both sexes".

Fig 1E legend - was testing performed separately in each cell type?

Yes, testing was performed separately on each cell type.

Fig 1H - it could be helpful to label a few selected "interesting" cell types on the left (I assume they were all identified in retrospect?). I also noticed that a couple of cell types express *dsx* and *fru* but do not show differences in abundance or gene expression - what are those?

We appreciate this suggestion, as some cell types only had small numbers of DEGs and little if any differential abundance, we have added a table reference to the statement in the results “While most central brain neuronal cell types displayed few or no DEGs between the sexes (Figure 1H, Table S1), specific cell types exhibited dramatic sex differences in the expression of individual genes.”. We have also added a supplemental figure (Figure S1) with these clusters annotated and references to the main text figures that further mention them.

Fig 2B, C - please explain how coverage was estimated.

Please see response to comment above.

Fig 2K legend - is the left graph males only?

We have now indicated that the graph on the left is sex-merged *dsx+* neuronal subtypes.

Fig 3D - does the dashed box correspond to Fig 3E?

We have now clarified this in the figure legend.

Fig 3H - why are some genes in bold? Transcription factors?

We have adjusted the legend of 3H: Temporal transcription factors identified over pseudotime are bolded (see 3I).

Fig 3I and Fig 4L - please mark early/late on the pseudotime bar (after that it should clear).

Done

Figure 5 - this would be an excellent place to introduce the background schematics of hemilineages and their timing.

Agreed, and we thank the reviewer for this helpful suggestion, which has now been added to Figure 5F-G.

Fig 6D - is there anything that defines subtype 7?

Yes, distinct markers for all subtypes are shown in 6C (e.g., *pdm3*, *Octbeta1R*, *AstC-R1*, *Drep3*).

Fig 6H legend - please mention what the arrowhead designates.

Our apologies, we have now added this designation to the figure legend.

Fig 7D-F - it's very hard to make sense of these data. All that can be seen is that some cell types express both early and late-born genes, while others are enriched for one or the other; and that most cell types clearly enriched for late or early genes are sexually monomorphic. The key point of how earliness/lateness relates to sexual dimorphism is not obvious from these figures. Could showing a more selective subset of the data help?

We appreciate that this figure was difficult to interpret and have made several updates to improve clarity. In addition to adding schematics to Figure 5 and clarifying the logic of birth order gene usage in the Results, we have made targeted revisions to Figure 7. Specifically, we now use a tSNE plot of the full central brain with Kenyon cells removed, whose atypical early/late gene profiles confounded interpretation (Figure 7D-F). We also added schematics to highlight the subset of neurons being analysed and their sexual dimorphism status (Figure 7D). Finally, we revised our schematic in Figure 7H to more clearly illustrate how birth order relates to sexual dimorphism across hemilineages.

Fig 7G - I found this way of representing the data incredibly confusing. I suspect a bar chart would be easier to interpret.

We thank the reviewer for this suggestion and have now used bar graphs (Figure 7G).

Line 1048 - I assume that, after these corrections, genes inferred to be sex-specific no longer exhibited strong dataset-specific expression - but this is not explicitly stated, and the post-correction data are not shown in Fig S2.

Apologies, there seems to be some confusion about what we have done. When fitting a model for the differential expression analyses, we included a covariate of dataset to attempt to regress out dataset-specific effects. However, this proved inadequate, so we further filtered out (i.e. removed genes) that met specific criteria suggesting that they were datasetspecific artifacts. The expression/counts from individual genes were not "corrected". These genes were then excluded from subsequent analyses, so there is no post-correction to be plotted. We have added the following text to the methods to clarify this: "Genes fitting these criteria were removed from all differential gene expression analyses shown in this manuscript". An example of a gene that did not show strong dataset-specific sex bias can be seen in Figure S3D.

Line 1071 - please explain what "l2-normalised" score means, and why it is being used.

We have now added an explanation of the l2-normalised score and why it was used in the Methods section "*Annotation of cell types and estimate of depth of coverage*".

Reviewer #2: The authors use single-cell transcriptomics to profile Fruitless and Doublesex expressing neurons throughout the central brain (excluding the optic lobes) - classifying sex-specific or sexually dimorphic cell types in males and females. This is an important study that complements recently released connectomes (hemibrain and FlyWire) for the female brain and a forthcoming male brain connectome (the comparisons of which can be used to define sex-specific and sexually dimorphic cell types based on connectivity, but connectome-based comparisons won't reveal differences in gene expression between cell types, what is covered here). Importantly, despite a number of studies on Fruitless and Doublesex expressing neurons in *Drosophila* over the past decade, we still know relatively little about the transcriptional differences that define Fru and Dsx cell types - this paper is an important advance towards that goal. The authors validate their findings by building split-GAL4 lines based on the results and showing that distinct combinations of gene expression define morphological cell types - these driver lines will be useful to the field. And, for some examples (like the Dh31-expressing mAL neurons), the authors make manipulations to determine why particular genes are expressed in male Fru neurons but not female Fru neurons. They identify two genes (*Imp* and *dati*) that mark early and late born neurons respectively, and relate this gene expression to female and male specific dsx/fru neurons, discovering that what sets the female brain apart from the male brain is when in development PCD (programmed cell death) occurs. Finally, the data are presented very well (figures are very easy to follow) and the writing is clear. I have very few concerns with the manuscript and am overall supportive of publication - my comments below should be addressed prior to publication.

We thank the reviewer for their thoughtful and supportive comments. We are particularly appreciative of the recognition of our efforts to link transcriptional identity with developmental timing and sex-specific neuronal diversity, and for highlighting how our atlas provides a critical and complementary perspective to connectome datasets. Below, we address the specific points raised.

Major Issues:

Fig. 1 - I could not find information on how neurons sequenced are sorted into cell types. Presumably there is some heterogeneity (both transcriptionally and in terms of synaptic connectivity (e.g., see Schlegel et al. and Matsliah et al. Nature 2024 papers - another, more exhaustive, way to define cell types)) within what is called a cell type here - how much? Cell types are critical to this paper - they are the fundamental unit used for making comparisons between male and female central brains - how they are defined (and what sources of error there are in defining them) needs to be laid out a bit more. Some of this may be presented in the companion paper (which I did not review) - would be helpful to a reader to have this information here.

We acknowledge that without reading the companion paper, these definitions are assumed. To address this, we have added a dedicated Methods section titled “**Annotation of cell types and estimate of depth of coverage**”, which details the strategy used to assign transcriptomic identities across the dataset. We have also revised the beginning of the Results section to include a clearer summary of the framework established in our companion paper, providing additional context for how we define and interpret cell types in this study.

Fig. 2 - see Deutsch et al. bioRxiv 2025 - there are 16 subtypes of Dsx+ neurons based on morphology and connectivity in the female brain (the number in the male brain is not yet known because the male brain connectome has not been released) - this study (Allen et al.) reports 6 subpopulations but refers to them as cell types. This is a bit confusing, and some explanation (again related to the definition of a cell type) would be helpful. As just one example, pC1 (covered in more depth in Fig. 6) is a cell type in this figure. But in female connectomes, pC1a-e are distinct cell types (again, based on morphology and connectivity) that drive different behaviors (e.g., receptivity versus aggression). Presumably pC1a-e have different transcriptional profiles - can those not be detected here (it looks like there might be clusters within the pC1 type in females in Fig 6)? If the 9.8x coverage is not high enough to detect differences between the pC1 subtypes (as just one example), make that clear. A similar issue also pertains to Fig. 3.

We acknowledge that the distinction between “cell type” and “subtype” can be confusing, particularly when trying to reconcile anatomical and transcriptional definitions. Throughout the manuscript, we now clarify our use of terminology. Specifically, for *dsx* and *fru* neurons, we reserve the term “cell type” to refer to transcriptionally defined populations that share a common hemilineage. Within these broad cell types, we observe considerable transcriptional heterogeneity, which we refer to as “subtypes”. Within *dsx* and *fru* cell types, we do have extensive transcriptional subtype diversity, which is seen in Figure S11. In Figure S11A-C, for example, we identify 28 pC1 transcriptional distinct subtypes. While we presume that the anatomically defined female subtypes (e.g., pC1a–e) correspond to subsets of these transcriptional clusters, direct alignment between these classifications was beyond the scope of this paper. Nevertheless, we provide subtype-specific marker genes in Figure S11C that could support future efforts to integrate transcriptional and connectomic datasets. We also note that although our dataset achieves 9.8× estimated coverage, it may still underrepresent rare subtypes, and additional sampling may be needed to resolve the full diversity observed in morphological reconstructions. This disclaimer has now been added to Methods section “Subclustering *doublesex* and *fruitless* neurons”.

Could not evaluate the underlying data (or comment on how it will be shared) at flycns.com - the website indicates it is 'coming soon'

We apologise for the confusion. The website does say "coming soon"; however, just beneath that is the hyperlink "Preview available here" where the interactive plotting of the full central brain neuronal atlas can be explored. We have now removed the "coming soon". All other analyses, with subclusterings, will go live upon publication.

Minor Issues/Other Comments:

Introduction - Line 69-70 does not make sense - the brain does not integrate sensory inputs, internal states, and behavioral outputs (or perhaps I don't understand the authors' meaning) - rephrase to indicate that the central brain of *Drosophila* (not including the optic lobes) integrates sensory information and drives behavioral outputs, both relative to internal states.

We thank the reviewer for pointing this out, the sentence has now been rephrased: “The central brain is an elaborately complex tissue critical for integrating sensory information and internal state to drive behavioral outputs.”

Line 103 - Can you justify the 9.8x coverage - what will be missed at this coverage level?

As stated above, this has now been added to the methods section “**Subclustering *doublesex* and *fruitless* neurons**”.

Line 138 - How is subtle defined? Given that the expectation is that only a fraction of the cell types in the central brain is sex-specific or sexually dimorphic, aren't the results consistent with that expectation? Also wouldn't the biggest differences in gene expression be during pupal development, not in adults?

This has now been updated and references added to clarify our intended meaning of subtle. “Our data support the prevailing model that sex influences the transcriptome of mature neurons through modest, cell-type-specific tuning within a largely shared framework, rather than through broad transcriptional reprogramming across the brain. This, however, does not rule out the possibility that more extensive sex-specific transcriptional differences may emerge earlier during development.”

Discussion - would be helpful to readers to discuss the results presented here in the context of published and emerging connectomes - connectivity is a complementary (and more exhaustive) way to define cell types - see for example Dombrowski et al. Nature 2025 or Yoo et al. CB 2023

We appreciate this thoughtful suggestion and agree that incorporating insights from connectomic data is an important future direction. To address this point, we have now added a sentence and references to the end of the Discussion highlighting how our transcriptomic atlas can complement emerging connectomic datasets. Specifically, we note that as complete adult brain connectomes become available for both sexes, our data will serve as a valuable resource for linking molecular and anatomical identities. We also clarify our view that transcriptomic identity provides a distinct and equally informative dimension for defining cell types. Integrating molecular and connectivity-based classifications will ultimately enable a more comprehensive understanding of the neural basis of sex-specific behaviours.

The authors report that Fru is expressed in 57% of hemilineages - what is special about the hemilineages that do not express Fru?

We thank the reviewer for raising this point. We have now directly compared Fru+ and Fru- hemilineages and did not identify any consistent transcriptional features that distinguish the two groups. At this time, we are unable to compare their anatomical properties based on our transcriptional dataset, as the anatomical identities of all fru+ hemilineages we identified have yet to be established. However, we agree this would be an interesting question to address in the future.

Referees' reports, second round of review

Reviewer #2: The authors have addressed all of my concerns - I am fully supportive of acceptance of the manuscript.